# Stochastic Cubic Regularization for Fast Nonconvex Optimization

**Nilesh Tripuraneni**\*   **Mitchell Stern**\*   **Chi Jin**   **Jeffrey Regier**   **Michael I. Jordan**
{nilesh_tripuraneni,mitchell,chijin,regier}@berkeley.edu
jordan@cs.berkeley.edu

University of California, Berkeley

## Abstract

This paper proposes a stochastic variant of a classic algorithm—the cubic-regularized Newton method [Nesterov and Polyak, 2006]. The proposed algorithm efficiently escapes saddle points and finds approximate local minima for general smooth, nonconvex functions in only $\tilde{\mathcal{O}}(\epsilon^{-3.5})$ stochastic gradient and stochastic Hessian-vector product evaluations. The latter can be computed as efficiently as stochastic gradients. This improves upon the $\tilde{\mathcal{O}}(\epsilon^{-4})$ rate of stochastic gradient descent. Our rate matches the best-known result for finding local minima without requiring any delicate acceleration or variance-reduction techniques.

## 1   Introduction

We consider the problem of nonconvex optimization in the stochastic approximation framework [Robbins and Monro, 1951]:

$$\min_{\mathbf{x} \in \mathbb{R}^d} f(\mathbf{x}) \coloneqq \mathbf{E}_{\xi \sim \mathcal{D}}[f(\mathbf{x}; \xi)]. \tag{1}$$

In this setting, we only have access to the stochastic function $f(\mathbf{x}; \xi)$, where the random variable $\xi$ is sampled from an underlying distribution $\mathcal{D}$. The task is to optimize the expected function $f(\mathbf{x})$, which in general may be nonconvex. This framework covers a wide range of problems, including the offline setting where we minimize the empirical loss over a fixed amount of data, and the online setting where data arrives sequentially. One of the most prominent applications of stochastic optimization has been in large-scale statistics and machine learning problems, such as the optimization of deep neural networks.

Classical analysis in nonconvex optimization only guarantees convergence to a first-order stationary point (i.e., a point $\mathbf{x}$ satisfying $\|\nabla f(\mathbf{x})\| = 0$), which can be a local minimum, a local maximum, or a saddle point. This paper goes further, proposing an algorithm that escapes saddle points and converges to a local minimum. A local minimum is defined as a point $\mathbf{x}$ satisfying $\|\nabla f(\mathbf{x})\| = 0$ and $\nabla^2 f(\mathbf{x}) \succeq 0$. Finding such a point is of special interest for a large class of statistical learning problems where local minima are global or near-global solutions (e.g. Choromanska et al. [2015], Sun et al. [2016a,b], Ge et al. [2017]).

Among first-order stochastic optimization algorithms, stochastic gradient descent (SGD) is perhaps the simplest and most versatile. While SGD is computationally inexpensive, the best current guarantee for finding an $\epsilon$-approximate local minimum (see Definition 1) requires $\mathcal{O}(\epsilon^{-4}\mathrm{poly}(d))$ iterations [Ge et al., 2015], which is inefficient in the high-dimensional regime.

---

In contrast, second-order methods which have access to the Hessian of $f$ can exploit curvature to more effectively escape saddles and arrive at local minima. However, constructing the full Hessian can be prohibitively expensive in high-dimensions. Thus, recent work has explored the use of Hessian-vector products $\nabla^2 f(\mathbf{x}) \cdot \mathbf{v}$, which can be computed as efficiently as gradients in many cases including neural networks [Pearlmutter, 1994].

Among second-order algorithms, one of the most natural extensions of the gradient descent algorithm is the cubic-regularized Newton method of Nesterov and Polyak [2006]. Whereas gradient descent finds the minimizer of a local second-order Taylor expansion at each step,

$$\mathbf{x}_{t+1}^{\text{GD}} = \operatorname*{argmin}_{\mathbf{x}} \left[ f(\mathbf{x}_t) + \nabla f(\mathbf{x}_t)^\top (\mathbf{x} - \mathbf{x}_t) + \frac{\ell}{2} \|\mathbf{x} - \mathbf{x}_t\|^2 \right],$$

the cubic regularized Newton method finds the minimizer of a local third-order Taylor expansion,

$$\mathbf{x}_{t+1}^{\text{Cubic}} = \operatorname*{argmin}_{\mathbf{x}} \left[ f(\mathbf{x}_t) + \nabla f(\mathbf{x}_t)^\top (\mathbf{x} - \mathbf{x}_t) + \frac{1}{2}(\mathbf{x} - \mathbf{x}_t)^\top \nabla^2 f(\mathbf{x}_t)(\mathbf{x} - \mathbf{x}_t) + \frac{\rho}{6}\|\mathbf{x} - \mathbf{x}_t\|^3 \right].$$

Most previous work on the cubic-regularized Newton method has focused on the non-stochastic or partially-stochastic settings. This leads us to ask the central questions of this paper: **Can we design a fully stochastic variant of the cubic-regularized Newton method? If so, is such an algorithm faster than SGD?**

In this work, we answer both questions in the affirmative, bridging the gap between its use in the non-stochastic and stochastic settings.

Specifically, we propose a stochastic variant of the cubic-regularized Newton method. We provide a non-asymptotic analysis of its complexity, showing that the proposed algorithm finds an $\epsilon$-second-order stationary point using only $\tilde{\mathcal{O}}(\epsilon^{-3.5})$ stochastic gradient and stochastic Hessian-vector evaluations, where $\tilde{\mathcal{O}}(\cdot)$ hides poly-logarithmic factors.[1] Our rate improves upon the $\tilde{\mathcal{O}}(\epsilon^{-4})$ rate of stochastic gradient descent, and matches the best-known result for finding local minima without the need for any delicate acceleration or variance reduction techniques (see Section 1.1 for details).

We also empirically show that the stochastic cubic-regularized Newton method proposed in this paper performs favorably on both synthetic and real non-convex problems relative to state-of-the-art optimization methods.

## 1.1 Related Work

There has been a recent surge of interest in optimization methods that can escape saddle points and find $\epsilon$-approximate local minima (see Definition 1) in various settings. We provide a brief summary of these results. All iteration complexities in this section are stated in terms of finding approximate local minima, and only highlight the dependency on $\epsilon$ and $d$.

### 1.1.1 Singleton Function

This line of work optimizes over a general function $f$ without any special structural assumptions. In this setting, the optimization algorithm has direct access to the gradient or Hessian oracles at each iteration. The work of Nesterov and Polyak [2006] first proposed the cubic-regularized Newton method, which requires $\mathcal{O}(\epsilon^{-1.5})$ gradient and Hessian oracle calls to the entire $f$, to find an $\epsilon$-second-order stationary point. Later, the ARC algorithm [Cartis et al., 2011] and trust-region methods [Curtis et al., 2017] were also shown to achieve the same guarantee with similar Hessian oracle access. However, these algorithms rely on having access to the full Hessian at each iteration, which is prohibitive in high dimensions.

Recently, instead of using the full Hessian, Carmon and Duchi [2016] showed that using a gradient descent solver for the cubic regularization subproblem allows their algorithm to find $\epsilon$-second-order stationary points in $\tilde{\mathcal{O}}(\epsilon^{-2})$ Hessian-vector product evaluations. With acceleration techniques, the number of Hessian-vector products can be reduced to $\tilde{\mathcal{O}}(\epsilon^{-1.75})$ [Carmon et al., 2016, Agarwal et al., 2017, Royer and Wright, 2017].

| Method | Runtime | Variance Reduction |
|---|---|---|
| Stochastic Gradient Descent [Ge et al., 2015] | $\mathcal{O}(\epsilon^{-4}\text{poly}(d))$ | not needed |
| Natasha 2 [Allen-Zhu, 2017] | $\tilde{\mathcal{O}}(\epsilon^{-3.5})^2$ | needed |
| **Stochastic Cubic Regularization (this paper)** | $\tilde{\mathcal{O}}(\epsilon^{-3.5})$ | not needed |

Table 1: Comparison of our results to existing results for stochastic, nonconvex optimization with provable convergence to approximate local minima.

Meanwhile, in the realm of entirely Hessian-free results, Jin et al. [2017] showed that a simple variant of gradient descent can find $\epsilon$-second stationary points in $\tilde{\mathcal{O}}(\epsilon^{-2})$ gradient evaluations.

Note this line of work doesn't accommodate stochastic gradients or stochastic Hessians. Restricting access to only stochastic function queries makes the optimization problem more challenging.

### 1.1.2 Finite-Sum Setting

In the finite-sum setting (also known as the offline setting) where $f(\mathbf{x}) := \frac{1}{n}\sum_{i=1}^{n} f_i(\mathbf{x})$, one assumes that algorithms have access to the individual functions $f_i$. In this setting, variance reduction techniques can be exploited [Johnson and Zhang, 2013]. Agarwal et al. [2017] give an algorithm requiring $\tilde{\mathcal{O}}(\frac{nd}{\epsilon^{3/2}} + \frac{n^{3/4}d}{\epsilon^{7/4}})$ Hessian-vector oracle calls (each to an $f_i(x)$) to find an $\epsilon$-approximate local minimum. A similar result is achieved by the algorithm proposed by Reddi et al. [2017].

### 1.1.3 Stochastic Approximation

The framework of stochastic approximation where $f(\mathbf{x}) := \mathbf{E}_{\xi \sim \mathcal{D}}[f(\mathbf{x}; \xi)]$ only assumes access to a stochastic gradient and Hessian via $f(\mathbf{x}; \xi)$. In this setting, Ge et al. [2015] showed that the total gradient iteration complexity for SGD to find an $\epsilon$-second-order stationary point was $\mathcal{O}(\epsilon^{-4}\text{poly}(d))$. More recently, Kohler and Lucchi [2017] consider a subsampled version of the cubic regularization algorithm, but do not provide a non-asymptotic analysis for their algorithm to find an approximate local minimum; they also assume access to exact (expected) function values at each iteration which are not available in the fully stochastic setting. Xu et al. [2017] consider the case of stochastic Hessians, but also require access to exact gradients and function values at each iteration. Recently, Allen-Zhu [2017] proposed an algorithm with a mechanism exploiting variance reduction that finds a second-order stationary point with $\tilde{\mathcal{O}}(\epsilon^{-3.5})^2$ Hessian-vector product evaluations.

After this work, Allen-Zhu and Li [2017] and Xu and Yang [2017] show how to use gradient evaluations to efficiently approximate Hessian-vector products. Using this technique along with variance reduction, they both provide algorithms achieving an $\tilde{\mathcal{O}}(\epsilon^{-3.5})$ rate using gradient evaluations.

We note that our result matches the best results so far, using a simpler approach without any delicate acceleration or variance-reduction techniques. See Table 1 for a brief comparison.

## 2 Preliminaries

We are interested in stochastic optimization problems of the form $\min_{\mathbf{x} \in \mathbb{R}^d} f(\mathbf{x}) := \mathbb{E}_{\xi \sim \mathcal{D}}[f(\mathbf{x}; \xi)]$, where $\xi$ is a random variable with distribution $\mathcal{D}$. In general, the function $f(\mathbf{x})$ may be nonconvex. This formulation covers both the standard offline setting where the objective function can be expressed as a finite sum of $n$ individual functions $f(\mathbf{x}, \xi_i)$, as well as the online setting where data arrives sequentially.

Our goal is to minimize the function $f(\mathbf{x})$ using only stochastic gradients $\nabla f(\mathbf{x}; \xi)$ and stochastic Hessian-vector products $\nabla^2 f(\mathbf{x}; \xi) \cdot \mathbf{v}$, where $\mathbf{v}$ is a vector of our choosing. Although it is expensive and often intractable in practice to form the entire Hessian, computing a Hessian-vector product is as cheap as computing a gradient when our function is represented as an arithmetic circuit [Pearlmutter, 1994], as is the case for neural networks.

**Notation**: We use bold uppercase letters $\mathbf{A}, \mathbf{B}$ to denote matrices and bold lowercase letters $\mathbf{x}, \mathbf{y}$ to denote vectors. For vectors we use $\|\cdot\|$ to denote the $\ell_2$-norm, and for matrices we use $\|\cdot\|$ to denote

the spectral norm and $\lambda_{\min}(\cdot)$ to denote the minimum eigenvalue. Unless otherwise specified, we use the notation $\mathcal{O}(\cdot)$ to hide only absolute constants which do not depend on any problem parameter, and the notation $\tilde{\mathcal{O}}(\cdot)$ to hide only absolute constants and logarithmic factors.

## 2.1 Assumptions

Throughout the paper, we assume that the function $f(\mathbf{x})$ is bounded below by some optimal value $f^*$. We also make the following assumptions about function smoothness:

**Assumption 1.** *The function $f(\mathbf{x})$ has*

- *$\ell$-Lipschitz gradients and $\rho$-Lipschitz Hessians: for all $\mathbf{x}_1$ and $\mathbf{x}_2$,*

$$\|\nabla f(\mathbf{x}_1) - \nabla f(\mathbf{x}_2)]\| \leq \ell\|\mathbf{x}_1 - \mathbf{x}_2\|; \left\|\nabla^2 f(\mathbf{x}_1) - \nabla^2 f(\mathbf{x}_2)\right\| \leq \rho\|\mathbf{x}_1 - \mathbf{x}_2\|.$$

The above assumptions state that the gradient and the Hessian cannot change dramatically in a small local area, and are standard in prior work on escaping saddle points and finding local minima.

Next, we make the following variance assumptions about stochastic gradients and stochastic Hessians:

**Assumption 2.** *The function $f(\mathbf{x}, \xi)$ has*

- *for all $\mathbf{x}$, $\mathbb{E}\left[\|\nabla f(\mathbf{x}, \xi) - \nabla f(\mathbf{x})\|^2\right] \leq \sigma_1^2$ and $\|\nabla f(\mathbf{x}, \xi) - \nabla f(\mathbf{x})\| \leq M_1$ a.s.;*
- *for all $\mathbf{x}$, $\mathbb{E}\left[\left\|\nabla^2 f(\mathbf{x}, \xi) - \nabla^2 f(\mathbf{x})\right\|^2\right] \leq \sigma_2^2$ and $\left\|\nabla^2 f(\mathbf{x}, \xi) - \nabla^2 f(\mathbf{x})\right\| \leq M_2$ a.s.*

We note that the above assumptions are not essential to our result, and can be replaced by any conditions that give rise to concentration. Moreover, the a.s. bounded Hessian assumption can be removed if one further assumes $f(\mathbf{x}, \xi)$ has $\ell'$-Lipschitz gradients for all $\xi$, which guarantees Hessian concentration with no further assumption on the variance of $\nabla^2 f(\mathbf{x}, \xi)$ [Tropp et al., 2015].

## 2.2 Cubic Regularization

Our target in this paper is to find an $\epsilon$-second-order stationary point, which we define as follows:

**Definition 1.** For a $\rho$-Hessian Lipschitz function $f$, we say that $\mathbf{x}$ is an *$\epsilon$-second-order stationary point* (or *$\epsilon$-approximate local minimum*) if

$$\|\nabla f(\mathbf{x})\| \leq \epsilon \quad \text{and} \quad \lambda_{\min}(\nabla^2 f(\mathbf{x})) \geq -\sqrt{\rho\epsilon}. \tag{2}$$

An $\epsilon$-second-order stationary point not only has a small gradient, but also has a Hessian which is close to positive semi-definite. Thus it is often also referred to as an $\epsilon$-approximate local minimum.

In the deterministic setting, cubic regularization [Nesterov and Polyak, 2006] is a classic algorithm for finding a second-order stationary point of a $\rho$-Hessian-Lipschitz function $f(\mathbf{x})$. In each iteration, it first forms a local upper bound on the function using a third-order Taylor expansion around the current iterate $\mathbf{x}_t$:

$$m_t(\mathbf{x}) = f(\mathbf{x}_t) + \nabla f(\mathbf{x}_t)^\top(\mathbf{x} - \mathbf{x}_t) + \frac{1}{2}(\mathbf{x} - \mathbf{x}_t)^\top\nabla^2 f(\mathbf{x}_t)(\mathbf{x} - \mathbf{x}_t) + \frac{\rho}{6}\|\mathbf{x} - \mathbf{x_t}\|^3.$$

This is called the *cubic submodel*. Then, cubic regularization minimizes this cubic submodel to obtain the next iterate: $\mathbf{x}_{t+1} = \operatorname{argmin}_{\mathbf{x}} m_t(\mathbf{x})$. When the cubic submodel can be solved exactly, cubic regularization requires $\mathcal{O}\left(\frac{\sqrt{\rho}(f(\mathbf{x}_0) - f^*)}{\epsilon^{1.5}}\right)$ iterations to find an $\epsilon$-second-order stationary point.

To apply this algorithm in the stochastic setting, three issues need to be addressed: (1) we only have access to stochastic gradients and Hessians, not the true gradient and Hessian; (2) our only means of interaction with the Hessian is through Hessian-vector products; (3) the cubic submodel cannot be solved exactly in practice, only up to some tolerance. We discuss how to overcome each of these obstacles in our paper.

## 3 Main Results

We begin with a general-purpose stochastic cubic regularization meta-algorithm in Algorithm 1, which employs a black-box subroutine to solve stochastic cubic subproblems. At a high level, in

---

**Algorithm 1** Stochastic Cubic Regularization (Meta-algorithm)

---

**Input:** mini-batch sizes $n_1, n_2$, initialization $\mathbf{x_0}$, number of iterations $T_{\text{out}}$, and final tolerance $\epsilon$.

1: **for** $t = 0, \ldots, T_{\text{out}}$ **do**
2:     Sample $S_1 \leftarrow \{\xi_i\}_{i=1}^{n_1}, S_2 \leftarrow \{\xi_i\}_{i=1}^{n_2}$.
3:     $\mathbf{g}_t \leftarrow \frac{1}{|S_1|} \sum_{\xi_i \in S_1} \nabla f(\mathbf{x}_t; \xi_i)$
4:     $\mathbf{B}_t[\cdot] \leftarrow \frac{1}{|S_2|} \sum_{\xi_i \in S_2} \nabla^2 f(\mathbf{x}_t, \xi_i)(\cdot)$
5:     $\boldsymbol{\Delta}, \Delta_m \leftarrow$ Cubic-Subsolver$(\mathbf{g}_t, \mathbf{B}_t[\cdot], \epsilon)$
6:     $\mathbf{x}_{t+1} \leftarrow \mathbf{x}_t + \boldsymbol{\Delta}$
7:     **if** $\Delta_m \geq -\frac{1}{100} \sqrt{\frac{\epsilon^3}{\rho}}$ **then**
8:         $\boldsymbol{\Delta} \leftarrow$ Cubic-Finalsolver$(\mathbf{g}_t, \mathbf{B}_t[\cdot], \epsilon)$
9:         $\mathbf{x}^* \leftarrow \mathbf{x}_t + \boldsymbol{\Delta}$
10:         **break**
11:     **end if**
12: **end for**

**Output:** $\mathbf{x}^*$ if the early termination condition was reached, otherwise the final iterate $x_{T_{\text{out}}+1}$.

---

order to deal with stochastic gradients and Hessians, we sample two independent minibatches $S_1$ and $S_2$ at each iteration. Denoting the average gradient and average Hessian by

$$\mathbf{g}_t = \frac{1}{|S_1|} \sum_{\xi_i \in S_1} \nabla f(\mathbf{x}_t, \xi_i), \mathbf{B}_t = \frac{1}{|S_2|} \sum_{\xi_i \in S_2} \nabla^2 f(\mathbf{x}_t, \xi_i), \tag{3}$$

this implies a *stochastic cubic submodel*:

$$m_t(\mathbf{x}) = f(\mathbf{x}_t) + (\mathbf{x} - \mathbf{x}_t)^\top \mathbf{g}_t + \frac{1}{2}(\mathbf{x} - \mathbf{x}_t)^\top \mathbf{B}_t(\mathbf{x} - \mathbf{x}_t) + \frac{\rho}{6}\|\mathbf{x} - \mathbf{x}_t\|^3. \tag{4}$$

Although the subproblem depends on $\mathbf{B}_t$, we note that our meta-algorithm never explicitly formulates this matrix, only providing the subsolver access to $\mathbf{B}_t$ through Hessian-vector products, which we denote by $\mathbf{B}_t[\cdot] : \mathbb{R}^d \to \mathbb{R}^d$. We hence assume that the subsolver performs gradient-based optimization to solve the subproblem, as $\nabla m_t(\mathbf{x})$ depends on $\mathbf{B}_t$ only via $\mathbf{B}_t[\mathbf{x} - \mathbf{x}_t]$.

After sampling minibatches for the gradient and the Hessian, Algorithm 1 makes a call to a black-box cubic subsolver to optimize the stochastic submodel $m_t(\mathbf{x})$. The subsolver returns a parameter change $\boldsymbol{\Delta}$, i.e., an approximate minimizer of the submodel, along with the corresponding change in submodel value, $\Delta_m := m_t(\mathbf{x}_t + \boldsymbol{\Delta}) - m_t(\mathbf{x}_t)$. The algorithm then updates the parameters by adding $\boldsymbol{\Delta}$ to the current iterate, and checks whether $\Delta_m$ satisfies a stopping condition.

In more detail, the Cubic-Subsolver subroutine takes a vector $\mathbf{g}$ and a function for computing Hessian-vector products $\mathbf{B}[\cdot]$, then optimizes the third-order polynomial $\tilde{m}(\boldsymbol{\Delta}) = \boldsymbol{\Delta}^\top \mathbf{g} + \frac{1}{2}\boldsymbol{\Delta}^\top \mathbf{B}[\boldsymbol{\Delta}] + \frac{\rho}{6}\|\boldsymbol{\Delta}\|^3$. Let $\boldsymbol{\Delta}^\star = \operatorname{argmin}_{\boldsymbol{\Delta}} \tilde{m}(\boldsymbol{\Delta})$ denote the minimizer of this polynomial. In general, the subsolver cannot return the exact solution $\boldsymbol{\Delta}^\star$. We hence tolerate a certain amount of suboptimality:

**Condition 1.** For any fixed, small constant $c$, Cubic-Subsolver$(\mathbf{g}, \mathbf{B}[\cdot], \epsilon)$ terminates within $\mathcal{T}(\epsilon)$ gradient iterations (which may depend on $c$), and returns a $\boldsymbol{\Delta}$ satisfying at least one of the following:

1. $\max\{\tilde{m}(\boldsymbol{\Delta}), f(\mathbf{x}_t + \boldsymbol{\Delta}) - f(\mathbf{x}_t)\} \leq -\Omega(\sqrt{\epsilon^3/\rho})$. (**Case 1**)

2. $\|\boldsymbol{\Delta}\| \leq \|\boldsymbol{\Delta}^\star\| + c\sqrt{\frac{\epsilon}{\rho}}$ and, if $\|\boldsymbol{\Delta}^\star\| \geq \frac{1}{2}\sqrt{\epsilon/\rho}$, then $\tilde{m}(\boldsymbol{\Delta}) \leq \tilde{m}(\boldsymbol{\Delta}^\star) + \frac{c}{12} \cdot \rho\|\boldsymbol{\Delta}^\star\|^3$. (**Case 2**)

The first condition is satisfied if the parameter change $\boldsymbol{\Delta}$ results in submodel and function decreases that are both sufficiently large (**Case 1**). If that fails to hold, the second condition ensures that $\boldsymbol{\Delta}$ is not too large relative to the true solution $\boldsymbol{\Delta}^\star$, and that the cubic submodel is solved to precision $c \cdot \rho\|\boldsymbol{\Delta}^\star\|^3$ when $\|\boldsymbol{\Delta}^\star\|$ is large (**Case 2**).

As mentioned above, we assume the subsolver uses gradient-based optimization to solve the subproblem so that it will only access the Hessian only through Hessian-vector products. Accordingly, it can be any standard first-order algorithm such as gradient descent, Nesterov's accelerated gradient

---

**Algorithm 2** Cubic-Subsolver via Gradient Descent

---

**Input:** $\mathbf{g}$, $\mathbf{B}[\cdot]$, tolerance $\epsilon$.

1: **if** $\|\mathbf{g}\| \geq \frac{\ell^2}{\rho}$ **then**

2: $\quad R_c \leftarrow -\frac{\mathbf{g}^\top \mathbf{B}[\mathbf{g}]}{\rho \|\mathbf{g}\|^2} + \sqrt{\left(\frac{\mathbf{g}^\top \mathbf{B}[\mathbf{g}]}{\rho \|\mathbf{g}\|^2}\right)^2 + \frac{2\|\mathbf{g}\|}{\rho}}$

3: $\quad \boldsymbol{\Delta} \leftarrow -R_c \frac{\mathbf{g}}{\|\mathbf{g}\|}$

4: **else**

5: $\quad \boldsymbol{\Delta} \leftarrow 0, \sigma \leftarrow c' \frac{\sqrt{\epsilon\rho}}{\ell}, \eta \leftarrow \frac{1}{20\ell}$

6: $\quad \tilde{\mathbf{g}} \leftarrow \mathbf{g} + \sigma\zeta$ for $\zeta \sim \text{Unif}(\mathbb{S}^{d-1})$

7: $\quad$ **for** $t = 1, \ldots, \mathcal{T}(\epsilon)$ **do**

8: $\quad\quad \boldsymbol{\Delta} \leftarrow \boldsymbol{\Delta} - \eta(\tilde{\mathbf{g}} + \mathbf{B}[\boldsymbol{\Delta}] + \frac{\rho}{2}\|\boldsymbol{\Delta}\|\boldsymbol{\Delta})$

9: $\quad$ **end for**

10: **end if**

11: $\Delta_m \leftarrow \mathbf{g}^\top \boldsymbol{\Delta} + \frac{1}{2}\boldsymbol{\Delta}^\top \mathbf{B}[\boldsymbol{\Delta}] + \frac{\rho}{6}\|\boldsymbol{\Delta}\|^3$

**Output:** $\boldsymbol{\Delta}, \Delta_m$

---

**Algorithm 3** Cubic-Finalsolver via Gradient Descent

---

**Input:** $\mathbf{g}$, $\mathbf{B}[\cdot]$, tolerance $\epsilon$.

1: $\boldsymbol{\Delta} \leftarrow 0, \mathbf{g}_m \leftarrow \mathbf{g}, \eta \leftarrow \frac{1}{20\ell}$

2: **while** $\|\mathbf{g}_m\| > \frac{\epsilon}{2}$ **do**

3: $\quad \boldsymbol{\Delta} \leftarrow \boldsymbol{\Delta} - \eta\mathbf{g}_m$

4: $\quad \mathbf{g}_m \leftarrow \mathbf{g} + \mathbf{B}[\boldsymbol{\Delta}] + \frac{\rho}{2}\|\boldsymbol{\Delta}\|\boldsymbol{\Delta}$

5: **end while**

**Output:** $\boldsymbol{\Delta}$

---

descent, etc. Gradient descent is of particular interest as it can be shown to satisfy Condition 1 (see Lemma 1).

Having given an overview of our meta-algorithm and verified the existence of a suitable subsolver, we are ready to present our main theorem:

**Theorem 1.** *There exists an absolute constant $c$ such that if $f(\mathbf{x})$ satisfies Assumptions 1, 2, CubicSubsolver satisfies Condition 1 with $c$, $n_1 \geq \max(\frac{M_1}{c\epsilon}, \frac{\sigma_1^2}{c^2\epsilon^2}) \log\left(\frac{d\sqrt{\rho}\Delta_f}{\epsilon^{1.5}\delta c}\right)$, and $n_2 \geq \max(\frac{M_2}{c\sqrt{\rho\epsilon}}, \frac{\sigma_2^2}{c^2\rho\epsilon}) \log\left(\frac{d\sqrt{\rho}\Delta_f}{\epsilon^{1.5}\delta c}\right)$, then for all $\delta > 0$ and $\Delta_f \geq f(\mathbf{x}_0) - f^*$, Algorithm 1 will output an $\epsilon$-second-order stationary point of $f$ with probability at least $1 - \delta$ within*

$$\tilde{\mathcal{O}}\left(\frac{\sqrt{\rho}\Delta_f}{\epsilon^{1.5}}\left(\max\left(\frac{M_1}{\epsilon}, \frac{\sigma_1^2}{\epsilon^2}\right) + \max\left(\frac{M_2}{\sqrt{\rho\epsilon}}, \frac{\sigma_2^2}{\rho\epsilon}\right) \cdot \mathcal{T}(\epsilon)\right)\right) \tag{5}$$

*total stochastic gradient and Hessian-vector product evaluations.*

In the limit where $\epsilon$ is small the result simplifies:

**Remark 1.** *If $\epsilon \leq \min\left\{\frac{\sigma_1^2}{c_1 M_1}, \frac{\sigma_2^4}{c_2^2 M_2^2 \rho}\right\}$, then under the settings of Theorem 1 we can conclude that Algorithm 1 will output an $\epsilon$-second-order stationary point of $f$ with probability at least $1 - \delta$ within*

$$\tilde{\mathcal{O}}\left(\frac{\sqrt{\rho}\Delta_f}{\epsilon^{1.5}}\left(\frac{\sigma_1^2}{\epsilon^2} + \frac{\sigma_2^2}{\rho\epsilon} \cdot \mathcal{T}(\epsilon)\right)\right) \tag{6}$$

total stochastic gradient and Hessian-vector product evaluations.

Theorem 1 states that after $T_{\text{out}} = \tilde{\mathcal{O}}(\frac{\sqrt{\rho}(f(\mathbf{x}_0) - f^*)}{\epsilon^{1.5}})$ iterations, stochastic cubic regularization (Algorithm 1) will have found an $\epsilon$-second-order stationary point w.h.p. Within each iteration, we require $n_1 = \tilde{\mathcal{O}}(\frac{\sigma_1^2}{\epsilon^2})$ samples for gradient averaging and $n_2 = \tilde{\mathcal{O}}(\frac{\sigma_2^2}{\rho\epsilon})$ samples for Hessian-vector product averaging. Recall that the averaged gradient $\mathbf{g}_t$ is fixed for a given cubic submodel, while the averaged Hessian-vector product $\mathbf{B}_t[\mathbf{v}]$ needs to be recalculated every time the cubic subsolver

queries the gradient $\nabla \tilde{m}(\cdot)$. At most $\mathcal{T}(\epsilon)$ such queries will be made by definition. Thus, each iteration takes $\tilde{\mathcal{O}}(\frac{\sigma_1^2}{\epsilon^2} + \frac{\sigma_2^2}{\rho\epsilon} \cdot \mathcal{T}(\epsilon))$ stochastic gradient/Hessian-vector product evaluations when $\epsilon$ is small (see Remark 1).

Finally, we note that lines 8-11 of Algorithm 1 give the termination condition of our meta-algorithm. When the decrease in submodel value $\Delta_m$ is too small, our theory guarantees $\mathbf{x}_t + \boldsymbol{\Delta}^\star$ is an $\epsilon$-second-order stationary point, where $\boldsymbol{\Delta}^\star$ is the optimal solution of the cubic submodel. However, Cubic-Subsolver may only produce an inexact solution $\boldsymbol{\Delta}$ satisfying Condition 1, which is not sufficient for $\mathbf{x}_t + \boldsymbol{\Delta}$ to be an $\epsilon$-second-order stationary point. We therefore call Cubic-Finalsolver (which just uses gradient descent and is detailed in Algorithm 3) to solve the subproblem with higher precision.

## 3.1 Gradient Descent as a Cubic-Subsolver

One concrete example of a cubic subsolver is a simple variant of gradient descent (Algorithm 2) as studied in Carmon and Duchi [2016]. The two main differences relative to standard gradient descent are: (1) lines 1–3: when $\mathbf{g}$ is large, the submodel $\tilde{m}(\boldsymbol{\Delta})$ may be ill-conditioned, so instead of doing gradient descent, the iterate only moves one step in the $\mathbf{g}$ direction, which already guarantees sufficient descent; (2) line 6: the algorithm adds a small perturbation to $\mathbf{g}$ to avoid a certain "hard" case for the cubic submodel. We refer readers to Carmon and Duchi [2016] for more details about Algorithm 2.

Adapting their result for our setting, we obtain the following lemma, which states that the gradient descent subsolver satisfies our Condition 1.

**Lemma 1.** *There exists an absolute constant $c'$, such that under the same assumptions on $f(\mathbf{x})$ and the same choice of parameters $n_1, n_2$ as in Theorem 1, Algorithm 2 satisfies Condition 1 with probability at least $1 - \delta'$ with $\mathcal{T}(\epsilon) \leq \tilde{\mathcal{O}}(\frac{\ell}{\sqrt{\rho\epsilon}})$.*

Our next corollary applies gradient descent (Algorithm 2) as the approximate cubic subsolver in our meta-algorithm (Algorithm 1), which immediately gives the total number of gradient and Hessian-vector evaluations for the full algorithm.

**Corollary 1.** *Under the same settings as Theorem 1, if $\epsilon \leq \min\left\{\frac{\sigma_1^2}{c_1 M_1}, \frac{\sigma_2^4}{c_2^2 M_2^2 \rho}\right\}$, and if we instantiate the Cubic-Subsolver subroutine with Algorithm 2, then with probability greater than $1 - \delta$, Algorithm 1 will output an $\epsilon$-second-order stationary point of $f(\mathbf{x})$ within*

$$\tilde{\mathcal{O}}\left(\frac{\sqrt{\rho}\Delta_f}{\epsilon^{1.5}}\left(\frac{\sigma_1^2}{\epsilon^2} + \frac{\sigma_2^2}{\rho\epsilon} \cdot \frac{\ell}{\sqrt{\rho\epsilon}}\right)\right) \tag{7}$$

*total stochastic gradient and Hessian-vector product evaluations.*

From Corollary 1, we observe that the dominant term in solving the submodel is $\frac{\sigma_1^2}{\epsilon^2}$ when $\epsilon$ is sufficiently small, giving a total iteration complexity of $\tilde{\mathcal{O}}(\epsilon^{-3.5})$ when other problem-dependent parameters are constant. This improves on the $\tilde{\mathcal{O}}(\epsilon^{-4}\text{poly}(d))$ complexity attained by SGD.

## 4 Proof Sketch

This section sketches the crucial steps needed to understand and prove our main theorem (Theorem 1). We describe our high-level approach, and provide a proof sketch in Appendix A in the stochastic setting, assuming oracle access to an exact subsolver. For the case of an inexact subsolver and full proof details, we defer to Appendix B.

Recall that at iteration $t$ of Algorithm 1, a stochastic cubic submodel $m_t$ is constructed around the current iterate $\mathbf{x}_t$ with the form given in the following Equation, $m_t(\mathbf{x}) = f(\mathbf{x}_t) + (\mathbf{x} - \mathbf{x}_t)^\top \mathbf{g}_t + \frac{1}{2}(\mathbf{x} - \mathbf{x}_t)^\top \mathbf{B}_t(\mathbf{x} - \mathbf{x}_t) + \frac{\rho}{6}\|\mathbf{x} - \mathbf{x}_t\|^3$, where $\mathbf{g}_t$ and $\mathbf{B}_t$ are the averaged stochastic gradients and Hessians. At a high level, we will show that for each iteration, the following two claims hold:

**Claim 1.** *If $\mathbf{x}_{t+1}$ is not an $\epsilon$-second-order stationary point of $f(\mathbf{x})$, the cubic submodel has large descent $m_t(\mathbf{x}_{t+1}) - m_t(\mathbf{x}_t)$.*

**Claim 2.** *If the cubic submodel has large descent $m_t(\mathbf{x}_{t+1}) - m_t(\mathbf{x}_t)$, then the true function also has large descent $f(\mathbf{x}_{t+1}) - f(\mathbf{x}_t)$.*

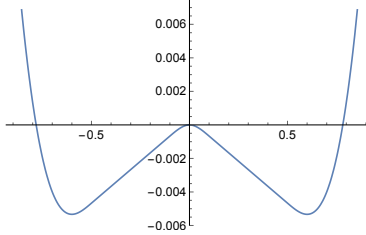

Figure 1: The piecewise cubic function $w(x)$ used along one of the dimensions in the synthetic experiment. The other dimension uses a scaled quadratic.

Given these claims, it is straightforward to argue for the correctness of our approach. We know that if we observe a large decrease in the cubic submodel value $m_t(\mathbf{x}_{t+1}) - m_t(\mathbf{x}_t)$ during Algorithm 1, then by Claim 2 the function will also have large descent. But since $f$ is bounded below, this cannot happen indefinitely, so we must eventually encounter an iteration with small cubic submodel descent. When that happens, we can conclude by Claim 1 that $\mathbf{x}_{t+1}$ is an $\epsilon$-second-order stationary point.

We note that Claim 2 is especially important in the stochastic setting, as we no longer have access to the true function but only the submodel. Claim 2 ensures that progress in $m_t$ still indicates progress in $f$, allowing the algorithm to terminate at the correct time. A more detailed proof sketch and full proofs can be found in the Appendix.

## 5   Experiments

In this section, we provide empirical results on synthetic and real-world data sets to demonstrate the efficacy of our approach. All experiments are implemented using TensorFlow [Abadi et al., 2016], which allows for efficient computation of Hessian-vector products using the method described by Pearlmutter [1994].

### 5.1   Synthetic Nonconvex Problem

We begin by constructing a nonconvex problem with a saddle point to compare our proposed approach against stochastic gradient descent. Let $w(x)$ be the W-shaped scalar function depicted in Figure 1, with a local maximum at the origin and two local minima on either side. We defer the exact form of $w(x)$ to Appendix D.

We aim to solve the problem

$$\min_{\mathbf{x} \in \mathbb{R}^2} \left[ w(x_1) + 10 x_2^2 \right],$$

with independent noise drawn from $\mathcal{N}(0, 1)$ added separately to each component of every gradient and Hessian-vector product evaluation. By construction, the objective function has a saddle point at the origin with Hessian eigenvalues of -0.2 and 20, providing a simple but challenging test case.

We ran our method and SGD on this problem, plotting the objective value versus the number of oracle calls in Figure 2. The batch sizes and learning rates for each method are tuned separately to ensure a fair comparison; see Appendix D for details. We observe that our method is able to escape the saddle point at the origin and converge to one of the global minima faster than SGD.

### 5.2   Deep Autoencoder

In addition to the synthetic problem above, we also investigate the performance of our approach on a more practical problem from deep learning, namely training a deep autoencoder on MNIST [LeCun and Cortes, 2010]. Our architecture consists of a fully connected encoder with dimensions $(28 \times 28) \rightarrow 512 \rightarrow 256 \rightarrow 128 \rightarrow 32$ together with a symmetric decoder. We use a softplus nonlinearity (defined as $\text{softplus}(x) = \log(1 + \exp(x))$) for each hidden layer, an elementwise sigmoid for the final layer, and a pixelwise $\ell_2$ loss between the input and the reconstructed output as our objective function. Results on this autoencoding task are presented in Figure 3. In addition to training the model with our method and SGD, we also include results using AdaGrad, a popular adaptive first-order method with strong empirical performance [Duchi et al., 2011], together with results for a method combining variance reduction and second-order information proposed by Reddi et al. [2017]. More details about our experimental setup can be found in Appendix D.

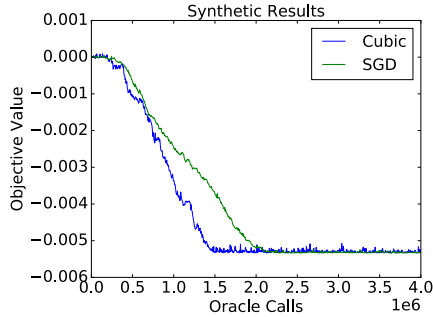

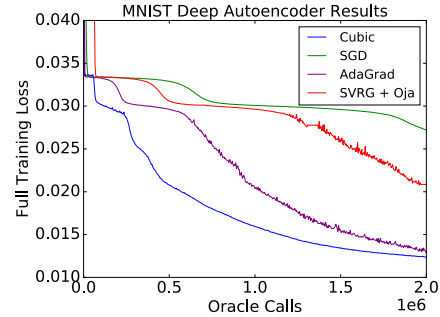

Figure 2: Results on our synthetic nonconvex optimization problem. Stochastic cubic regularization escapes the saddle point at the origin and converges to a global optimum faster than SGD.

Figure 3: Results on the MNIST deep autoencoding task. Multiple saddle points are present in the optimization problem. Stochastic cubic regularization is able to escape them most quickly, allowing it to reach a local minimum faster than other methods.

We observe that stochastic cubic regularization quickly escapes two saddle points and descends toward a local optimum, while AdaGrad takes two to three times longer to escape each saddle point, and SGD is slower still. This demonstrates that incorporating curvature information via Hessian-vector products can assist in escaping saddle points in practice. The hybrid method consisting of SVRG interleaved with Oja's algorithm for Hessian descent also improves upon SGD but does not match the performance of our method or AdaGrad.

## 6   Conclusion

In this paper, we presented a stochastic algorithm based on the classic cubic-regularized Newton method for nonconvex optimization. Our algorithm provably finds $\epsilon$-approximate local minima in $\tilde{\mathcal{O}}(\epsilon^{-3.5})$ total gradient and Hessian-vector product evaluations, improving upon the $\tilde{\mathcal{O}}(\epsilon^{-4}\text{poly}(d))$ rate of SGD. Our experiments show the favorable performance of our method relative to SGD on both a synthetic and a real-world problem.

## Footnotes

[1]A single query, for a single realization $\xi \sim \mathcal{D}$, to $\nabla f(\mathbf{x}, \xi)$ or $\nabla^2 f(\mathbf{x}, \xi) \cdot \mathbf{v}$ (for pre-specified $\mathbf{v}$) is referred to as a stochastic gradient or stochastic Hessian-vector oracle evaluation.

[2]The original paper reports a rate of $\tilde{\mathcal{O}}(\epsilon^{-3.25})$ due to a different definition of $\epsilon$-second-order stationary point, $\lambda_{\min}(\nabla^2 f(x)) \geq -\mathcal{O}(\epsilon^{1/4})$, which is weaker than the standard definition as in Definition 1.

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
