[Supplementary Material]

# A Proof Sketch

Before providing complete proofs of our main results we provide a brief proof sketch to motivate the approach.

## A.1 Stochastic Setting with Exact Subsolver

Recall that at iteration $t$ of Algorithm 1, a stochastic cubic submodel $m_t$ is constructed around the current iterate $\mathbf{x}_t$ with the form given in the following Equation, $m_t(\mathbf{x}) = f(\mathbf{x}_t) + (\mathbf{x} - \mathbf{x}_t)^\top \mathbf{g}_t + \frac{1}{2}(\mathbf{x} - \mathbf{x}_t)^\top \mathbf{B}_t(\mathbf{x} - \mathbf{x}_t) + \frac{\rho}{6}\|\mathbf{x} - \mathbf{x}_t\|^3$, where $\mathbf{g}_t$ and $\mathbf{B}_t$ are the averaged stochastic gradients and Hessians.

In this setting, $\mathbf{g}_t$ and $\mathbf{B}_t$ are the averaged gradient and Hessian with sample sizes $n_1$ and $n_2$, respectively. To ensure the stochastic cubic submodel approximates the exact cubic submodel well, we need large enough sample sizes so that both $\mathbf{g}_t$ and $\mathbf{B}_t$ are close to the exact gradient and Hessian at $\mathbf{x}_t$ up to some tolerance:

**Lemma 2.** *For any fixed small constants $c_1, c_2$, we can pick gradient and Hessian mini-batch sizes $n_1 = \tilde{\mathcal{O}}\left(\max\left(\frac{M_1}{\epsilon}, \frac{\sigma_1^2}{\epsilon^2}\right)\right)$ and $n_2 = \tilde{\mathcal{O}}\left(\max\left(\frac{M_2}{\sqrt{\rho\epsilon}}, \frac{\sigma_2^2}{\rho\epsilon}\right)\right)$ so that with probability $1 - \delta'$,*

$$\|\mathbf{g}_t - \nabla f(\mathbf{x}_t)\| \leq c_1 \cdot \epsilon, \tag{8}$$

$$\forall \mathbf{v}, \|(\mathbf{B}_t - \nabla^2 f(\mathbf{x}_t))\mathbf{v}\| \leq c_2 \cdot \sqrt{\rho\epsilon}\|\mathbf{v}\|. \tag{9}$$

We need to ensure that the random vectors/matrices concentrate along an arbitrary direction (depending on $\mathbf{g}_t$ and $\mathbf{B}_t$). In order to guarantee the uniform concentration in Lemma 2, we can directly apply results from matrix concentration to obtain the desired result [Tropp et al., 2015].

Let $\mathbf{\Delta}_t^\star = \text{argmin}_{\mathbf{\Delta}} \, m_t(\mathbf{x}_t + \mathbf{\Delta})$, i.e. $\mathbf{x}_t + \mathbf{\Delta}_t^\star$ is a global minimizer of the cubic submodel $m_t$. If we use an exact oracle solver, we have $\mathbf{x}_{t+1} = \mathbf{x}_t + \mathbf{\Delta}_t^\star$. In order to show Claim 1 and Claim 2, one important quantity to study is the decrease in the cubic submodel $m_t$:

**Lemma 3.** *Let $m_t$ and $\mathbf{\Delta}_t^\star$ be defined as above. Then for all $t$,*

$$m_t(\mathbf{x}_t + \mathbf{\Delta}_t^\star) - m_t(\mathbf{x}_t) \leq -\frac{1}{12}\rho\|\mathbf{\Delta}_t^\star\|^3.$$

Lemma 3 implies that in order to prove submodel $m_t$ has sufficient function value decrease, we only need to lower bound the norm of optimal solution, i.e. $\|\mathbf{\Delta}_t^\star\|$.

**Proof sketch for claim 1:** Our strategy is to lower bound the norm of $\mathbf{\Delta}_t^\star$ when $\mathbf{x}_{t+1} = \mathbf{x}_t + \mathbf{\Delta}_t^\star$ is not an $\epsilon$-second-order stationary point. In the non-stochastic setting, Nesterov and Polyak [2006] prove

$$\|\mathbf{\Delta}_t^\star\| \geq \frac{1}{2}\max\left\{\sqrt{\frac{1}{\rho}\|\nabla f(\mathbf{x}_{t+1})\|}, \frac{1}{\rho}\lambda_{\min}(\nabla^2 f(\mathbf{x}_{t+1}))\right\},$$

which gives the desired result. In the stochastic setting, a similar statement holds up to some tolerance:

**Lemma 4.** *Under the setting of Lemma 2 with sufficiently small constants $c_1, c_2$,*

$$\|\mathbf{\Delta}_t^\star\| \geq \frac{1}{2}\max\left\{\sqrt{\frac{1}{\rho}\left(\|\nabla f(\mathbf{x}_t + \mathbf{\Delta}_t^\star)\| - \frac{\epsilon}{4}\right)}, \frac{1}{\rho}\left(\lambda_{\min}(\nabla^2 f(\mathbf{x}_t + \mathbf{\Delta}_t^\star)) - \frac{\sqrt{\rho\epsilon}}{4}\right)\right\}.$$

That is, when $\mathbf{x}_{t+1}$ is not an $\epsilon$-second-order stationary point, we have $\|\mathbf{\Delta}_t^\star\| \geq \Omega(\sqrt{\frac{\epsilon}{\rho}})$. In other words, we have sufficient movement. It follows by Lemma 3 that we have sufficient cubic submodel descent.

**Proof sketch for claim 2:** In the non-stochastic case, $m_t(\mathbf{x})$ is by construction an upper bound on $f(\mathbf{x})$. Together with the fact $f(\mathbf{x}_t) = m_t(\mathbf{x}_t)$, we have:

$$f(\mathbf{x}_{t+1}) - f(\mathbf{x}_t) \leq m_t(\mathbf{x}_{t+1}) - m_t(\mathbf{x}_t),$$

showing Claim 2 is always true. For the stochastic case, this inequality may no longer be true. Instead, under the setting of Lemma 2, via Lemma 3, we can upper bound the function decrease with an additional error term:

$$f(\mathbf{x}_{t+1}) - f(\mathbf{x}_t) \leq \frac{1}{2}[m_t(\mathbf{x}_{t+1}) - m_t(\mathbf{x}_t)] + c\sqrt{\frac{\epsilon^3}{\rho}},$$

for some sufficiently small constant $c$. Then when $m_t(\mathbf{x}_{t+1}) - m_t(\mathbf{x}_t) \leq -4c\sqrt{\epsilon^3/\rho}$, we have $f(\mathbf{x}_{t+1}) - f(\mathbf{x}_t) \leq \frac{1}{4}[m_t(\mathbf{x}_{t+1}) - m_t(\mathbf{x}_t)] \leq -c\sqrt{\epsilon^3/\rho}$, which proves Claim 2.

Finally, for an approximate cubic subsolver, the story becomes more elaborate. Claim 1 is only "approximately" true, while Claim 2 still holds but for more complicated reasons.

# B Proof of Main Results

In this section, we give formal proofs of Theorems 1 and Corollary 1. We start by providing proofs of several useful auxiliary lemmas.

**Remark 2.** It suffices to assume that $\epsilon \leq \frac{\ell^2}{\rho}$ for the following analysis, since otherwise every point $\mathbf{x}$ satisfies the second-order condition $\lambda_{\min}(\nabla^2 f(\mathbf{x})) \geq -\sqrt{\rho\epsilon}$ trivially by the Lipschitz-gradient assumption.

## B.1 Set-Up and Notation

Here we remind the reader of the relevant notation and provide further background from Nesterov and Polyak [2006] on the cubic-regularized Newton method. We denote the stochastic gradient as

$$\mathbf{g}_t = \frac{1}{|S_1|} \sum_{\xi_i \in S_1} \nabla f(\mathbf{x}_t, \xi_i)$$

and the stochastic Hessian as

$$\mathbf{B}_t = \frac{1}{|S_2|} \sum_{\xi_i \in S_2} \nabla^2 f(\mathbf{x}_t, \xi_i),$$

both for iteration $t$. We draw a sufficient number of samples $|S_1|$ and $|S_2|$ so that the concentration conditions

$$\|\mathbf{g}_t - \nabla f(\mathbf{x}_t)\| \leq c_1 \cdot \epsilon,$$
$$\forall \mathbf{v}, \|(\mathbf{B}_t - \nabla^2 f(\mathbf{x}_t))\mathbf{v}\| \leq c_2 \cdot \sqrt{\rho\epsilon}\|\mathbf{v}\|.$$

are satisfied for sufficiently small $c_1, c_2$ (see Lemma 2 for more details). The cubic-regularized Newton subproblem is to minimize

$$m_t(\mathbf{y}) = f(\mathbf{x}_t) + (\mathbf{y} - \mathbf{x}_t)^\top \mathbf{g}_t + \frac{1}{2}(\mathbf{y} - \mathbf{x}_t)^\top \mathbf{B}_t(\mathbf{y} - \mathbf{x}_t) + \frac{\rho}{6}\|\mathbf{y} - \mathbf{x}_t\|^3. \quad (10)$$

We denote the global optimizer of $m_t(\cdot)$ as $\mathbf{x}_t + \mathbf{\Delta}_t^\star$, that is $\mathbf{\Delta}_t^\star = \operatorname{argmin}_z m_k(\mathbf{z} + \mathbf{x}_k)$.

As shown in Nesterov and Polyak [2006] a global optimum of Equation (10) satisfies:

$$\mathbf{g}_t + \mathbf{B}_t\mathbf{\Delta}_t^\star + \frac{\rho}{2}\|\mathbf{\Delta}_t^\star\|\mathbf{\Delta}_t^\star = 0. \quad (11)$$

$$\mathbf{B}_t + \frac{\rho}{2}\|\mathbf{\Delta}_t^\star\|I \succeq 0. \quad (12)$$

Equation (11) is the first-order stationary condition, while Equation (12) follows from a duality argument. In practice, we will not be able to directly compute $\mathbf{\Delta}_t^\star$ so will instead use a Cubic-Subsolver routine which must satisfy:

**Condition 1.** For any fixed, small constant $c_3, c_4$, Cubic-Subsolver$(\mathbf{g}, \mathbf{B}[\cdot], \epsilon)$ terminates within $\mathcal{T}(\epsilon)$ gradient iterations (which may depend on $c_3, c_4$), and returns a $\mathbf{\Delta}$ satisfying at least one of the following:

1. $\max\{\tilde{m}(\mathbf{\Delta}), f(\mathbf{x}_t + \mathbf{\Delta}) - f(\mathbf{x}_t)\} \leq -\Omega(\sqrt{\epsilon^3/\rho})$. **(Case 1)**

2. $\|\mathbf{\Delta}\| \leq \|\mathbf{\Delta}^\star\| + c_4\sqrt{\frac{\epsilon}{\rho}}$ and, if $\|\mathbf{\Delta}^\star\| \geq \frac{1}{2}\sqrt{\epsilon/\rho}$, then $\tilde{m}(\mathbf{\Delta}) \leq \tilde{m}(\mathbf{\Delta}^\star) + \frac{c_3}{12} \cdot \rho\|\mathbf{\Delta}^\star\|^3$.
   **(Case 2)**

## B.2 Auxiliary Lemmas

We begin by providing the proof of several useful auxiliary lemmas. First we provide the proof of Lemma 2 which characterize the concentration conditions.

**Lemma 2.** *For any fixed small constants $c_1, c_2$, we can pick gradient and Hessian mini-batch sizes $n_1 = \tilde{\mathcal{O}}\left(\max\left(\frac{M_1}{\epsilon}, \frac{\sigma_1^2}{\epsilon^2}\right)\right)$ and $n_2 = \tilde{\mathcal{O}}\left(\max\left(\frac{M_2}{\sqrt{\rho\epsilon}}, \frac{\sigma_2^2}{\rho\epsilon}\right)\right)$ so that with probability $1 - \delta'$,*

$$\|\mathbf{g}_t - \nabla f(\mathbf{x}_t)\| \leq c_1 \cdot \epsilon, \tag{8}$$

$$\forall \mathbf{v}, \|(\mathbf{B}_t - \nabla^2 f(\mathbf{x}_t))\mathbf{v}\| \leq c_2 \cdot \sqrt{\rho\epsilon}\|\mathbf{v}\|. \tag{9}$$

*Proof.* We can use the matrix Bernstein inequality from Tropp et al. [2015] to control both the fluctuations in the stochastic gradients and stochastic Hessians under Assumption 2.

Recall that the spectral norm of a vector is equivalent to its vector norm. So the matrix variance of the centered gradients $\tilde{\mathbf{g}} = \frac{1}{n_1}\sum_{i=1}^{n_1}\left(\tilde{\nabla}f(\mathbf{x}, \xi_i)\right) = \frac{1}{n_1}\sum_{i=1}^{n_1}\left(\nabla f(\mathbf{x}, \xi_i) - \nabla f(\mathbf{x})\right)$ is:

$$v[\tilde{\mathbf{g}}] = \frac{1}{n_1^2}\max\left\{\left\|\mathbb{E}\left[\sum_{i=1}^{n_1}\tilde{\nabla}f(\mathbf{x}, \xi_i)\tilde{\nabla}f(\mathbf{x}, \xi_i)^\top\right]\right\|, \left\|\mathbb{E}\left[\sum_{i=1}^{n_1}\tilde{\nabla}f(\mathbf{x}, \xi_i)^\top\tilde{\nabla}f(\mathbf{x}, \xi_i)\right]\right\|\right\} \leq \frac{\sigma_1^2}{n_1}$$

using the triangle inequality and Jensen's inequality. A direct application of the matrix Bernstein inequality gives:

$$\mathbb{P}\left[\|\mathbf{g} - \nabla f(\mathbf{x})\| \geq t\right] \leq 2d\exp\left(-\frac{t^2/2}{v[\tilde{\mathbf{g}}] + M_1/(3n_1)}\right) \leq 2d\exp\left(-\frac{3n_1}{8}\min\left\{\frac{t}{M_1}, \frac{t^2}{\sigma_1^2}\right\}\right) \implies$$

$$\|\mathbf{g} - \nabla f(\mathbf{x})\| \leq t \text{ with probability } 1 - \delta' \text{ for } n_1 \geq \max\left(\frac{M_1}{t}, \frac{\sigma_1^2}{t^2}\right)\frac{8}{3}\log\frac{2d}{\delta'}$$

Taking $t = c_1\epsilon$ gives the result.

The matrix variance of the centered Hessians $\tilde{\mathbf{B}} = \frac{1}{n_2}\sum_{i=1}^{n_2}\left(\tilde{\nabla}^2 f(\mathbf{x}, \xi_i)\right) = \frac{1}{n_2}\sum_{i=1}^{n_2}\left(\nabla^2 f(\mathbf{x}, \xi_i) - \nabla^2 f(\mathbf{x})\right)$, which are symmetric, is:

$$v[\tilde{\mathbf{B}}] = \frac{1}{n_2^2}\left\|\sum_{i=1}^{n_2}\mathbb{E}\left[\left(\tilde{\nabla}^2 f(\mathbf{x}, \xi_i)\right)^2\right]\right\| \leq \frac{\sigma_2^2}{n_2} \tag{13}$$

once again using the triangle inequality and Jensen's inequality. Another application of the matrix Bernstein inequality gives that:

$$\mathbb{P}[\|\mathbf{B} - \nabla^2 f(\mathbf{x}))\| \geq t] \leq 2d\exp\left(-\frac{3n_2}{8}\min\{\frac{t}{M_2}, \frac{t^2}{\sigma_2^2}\}\right) \implies$$

$$\left\|\mathbf{B} - \nabla^2 f(\mathbf{x}))\right\| \leq t \text{ with probability } 1 - \delta' \text{ for } n_2 \geq \max(\frac{M_2}{t}, \frac{\sigma_2^2}{t^2})\frac{8}{3}\log\frac{2d}{\delta'}$$

Taking $t = c_2\sqrt{\rho\epsilon}$ ensures that the stochastic Hessian-vector products are controlled uniformly over $\mathbf{v}$:

$$\left\|(\mathbf{B} - \nabla^2 f(\mathbf{x}))\mathbf{v}\right\| \leq c_2 \cdot \sqrt{\rho\epsilon}\|\mathbf{v}\|$$

using $n_2$ samples with probability $1 - \delta'$.

$\square$

Next we show Lemma 3 which will relate the change in the cubic function value to the norm $\|\mathbf{\Delta}_t^\star\|$.

**Lemma 3.** *Let $m_t$ and $\mathbf{\Delta}_t^\star$ be defined as above. Then for all $t$,*

$$m_t(\mathbf{x}_t + \mathbf{\Delta}_t^\star) - m_t(\mathbf{x}_t) \leq -\frac{1}{12}\rho\|\mathbf{\Delta}_t^\star\|^3.$$

*Proof.* Using the global optimality conditions for Equation (10) from Nesterov and Polyak [2006], we have the global optimum $\mathbf{x}_t + \boldsymbol{\Delta}_t^\star$, satisfies:

$$\mathbf{g}_t + \mathbf{B}_t(\boldsymbol{\Delta}_t^\star) + \frac{\rho}{2}\|\boldsymbol{\Delta}_t^\star\|(\boldsymbol{\Delta}_t^\star) = 0$$

$$\mathbf{B}_t + \frac{\rho}{2}\|\boldsymbol{\Delta}_t^\star\|I \succeq 0.$$

Together these conditions also imply that:

$$(\boldsymbol{\Delta}_t^\star)^\top \mathbf{g}_t + (\boldsymbol{\Delta}_t^\star)^\top \mathbf{B}_t(\boldsymbol{\Delta}_t^\star) + \frac{\rho}{2}\|\boldsymbol{\Delta}_t^\star\|^3 = 0$$

$$(\boldsymbol{\Delta}_t^\star)^\top \mathbf{B}_t(\boldsymbol{\Delta}_t^\star) + \frac{\rho}{2}\|\boldsymbol{\Delta}_t^\star\|^3 \geq 0.$$

Now immediately from the definition of stochastic cubic submodel model and the aforementioned conditions we have that:

$$f(\mathbf{x}_t) - m_t(\mathbf{x}_t + \boldsymbol{\Delta}_t^\star) = -(\boldsymbol{\Delta}_t^\star)^\top \mathbf{g}_t - \frac{1}{2}(\boldsymbol{\Delta}_t^\star)^\top \mathbf{B}_t(\boldsymbol{\Delta}_t^\star) - \frac{\rho}{6}\|\mathbf{x}_t + \boldsymbol{\Delta}_t^\star\|^3$$

$$= \frac{1}{2}(\boldsymbol{\Delta}_t^\star)^\top \mathbf{B}_t(\boldsymbol{\Delta}_t^\star) + \frac{1}{3}\rho\|\boldsymbol{\Delta}_t^\star\|^3$$

$$\geq \frac{1}{12}\rho\|\boldsymbol{\Delta}_t^\star\|^3$$

An identical statement appears as Lemma 10 in Nesterov and Polyak [2006], so this is merely restated here for completeness. $\square$

Thus to guarantee sufficient descent it suffices to lower bound the $\|\boldsymbol{\Delta}_t^\star\|$. We now prove Lemma 4, which guarantees the sufficient "movement" for the exact update: $\|\boldsymbol{\Delta}_t^\star\|$. In particular this will allow us to show that when $\mathbf{x}_t + \boldsymbol{\Delta}_t^\star$ is not an $\epsilon$-second-order stationary point then $\|\boldsymbol{\Delta}_t^\star\| \geq \frac{1}{2}\sqrt{\frac{\epsilon}{\rho}}$.

**Lemma 4.** *Under the setting of Lemma 2 with sufficiently small constants* $c_1, c_2$,

$$\|\boldsymbol{\Delta}_t^\star\| \geq \frac{1}{2}\max\left\{\sqrt{\frac{1}{\rho}\left(\|\nabla f(\mathbf{x}_t + \boldsymbol{\Delta}_t^\star)\| - \frac{\epsilon}{4}\right)}, \frac{1}{\rho}\left(\lambda_{\min}(\nabla^2 f(\mathbf{x}_t + \boldsymbol{\Delta}_t^\star)) - \frac{\sqrt{\rho\epsilon}}{4}\right)\right\}.$$

*Proof.* As a consequence of the global optimality condition, given in Equation (11), we have that:

$$\|\mathbf{g}_t + \mathbf{B}_t(\boldsymbol{\Delta}_t^\star)\| = \frac{\rho}{2}\|\boldsymbol{\Delta}_t^\star\|^2. \tag{14}$$

Moreover, from the Hessian-Lipschitz condition it follows that:

$$\left\|\nabla f(\mathbf{x}_t + \boldsymbol{\Delta}_t^\star) - \nabla f(\mathbf{x}_t) - \nabla^2 f(\mathbf{x}_t)(\boldsymbol{\Delta}_t^\star)\right\| \leq \frac{\rho}{2}\|\boldsymbol{\Delta}_t^\star\|^2. \tag{15}$$

Combining the concentration assumptions with Equation (14) and Inequality (15), we obtain:

$$\|\nabla f(\mathbf{x}_t + \boldsymbol{\Delta}_t^\star)\| = \left\|\nabla f(\mathbf{x}_t + \boldsymbol{\Delta}_t^\star) - \nabla f(\mathbf{x}_t) - \nabla^2 f(\mathbf{x}_t)(\boldsymbol{\Delta}_t^\star)\right\| + \left\|\nabla f(\mathbf{x}_t) + \nabla^2 f(\mathbf{x}_t)(\boldsymbol{\Delta}_t^\star)\right\|$$

$$\leq \left\|\nabla f(\mathbf{x}_t + \boldsymbol{\Delta}_t^\star) - \nabla f(\mathbf{x}_t) - \nabla^2 f(\mathbf{x}_t)(\boldsymbol{\Delta}_t^\star)\right\| + \|\mathbf{g}_t + \mathbf{B}_t(\boldsymbol{\Delta}_t^\star)\|$$

$$+ \|\mathbf{g}_t - \nabla f(\mathbf{x}_t)\| + \left\|(\mathbf{B}_t - \nabla^2 f(\mathbf{x}_t))\boldsymbol{\Delta}_t^\star\right\|$$

$$\leq \rho\|\boldsymbol{\Delta}_t^\star\|^2 + c_1\epsilon + c_2\sqrt{\rho\epsilon}\|\boldsymbol{\Delta}_t^\star\|. \tag{16}$$

An application of the Fenchel-Young inequality to the final term in Equation (16) then yields:

$$\|\nabla f(\mathbf{x}_t + \boldsymbol{\Delta}_t^\star)\| \leq \rho(1 + \frac{c_2}{2})\|\boldsymbol{\Delta}_t^\star\|^2 + (c_1 + \frac{c_2}{2})\epsilon \implies$$

$$\frac{1}{\rho(1 + \frac{c_2}{2})}\left(\|\nabla f(\mathbf{x}_t + \boldsymbol{\Delta}_t^\star)\| - (c_1 + \frac{c_2}{2})\epsilon\right) \leq \|\boldsymbol{\Delta}_t^\star\|^2,$$

which lower bounds $\|\boldsymbol{\Delta}_t^\star\|$ with respect to the gradient at $\mathbf{x}_t$. For the corresponding Hessian lower bound we first utilize the Hessian Lipschitz condition:

$$\nabla^2 f(\mathbf{x}_t + \boldsymbol{\Delta}_t^\star) \succeq \nabla^2 f(\mathbf{x}_t) - \rho\|\boldsymbol{\Delta}_t^\star\|I$$

$$\succeq \mathbf{B}_t - c_2\sqrt{\rho\epsilon}I - \rho\|\boldsymbol{\Delta}_t^\star\|I$$
$$\succeq -c_2\sqrt{\rho\epsilon}I - \frac{3}{2}\rho\|\boldsymbol{\Delta}_t^\star\|I,$$

followed by the concentration condition and the optimality condition (12). This immediately implies

$$\|\boldsymbol{\Delta}_t^\star\|I \succeq -\frac{2}{3\rho}\left(\nabla^2 f(\mathbf{x}_t + \boldsymbol{\Delta}_t^\star) + c_2\sqrt{\rho\epsilon}I\right) \implies$$
$$\|\boldsymbol{\Delta}_t^\star\| \geq -\frac{2}{3\rho}\lambda_{\min}(\nabla^2 f(\mathbf{x}_t + \boldsymbol{\Delta}_t^\star)) - \frac{2c_2}{3\sqrt{\rho}}\sqrt{\epsilon}$$

Combining we obtain that:

$$\|\boldsymbol{\Delta}_t^\star\| \geq \max\left\{\sqrt{\frac{1}{\rho(1 + \frac{c_2}{2})}\left(\|\nabla f(\mathbf{x}_t + \boldsymbol{\Delta}_t^\star)\| - (c_1 + \frac{c_2}{2})\epsilon\right)}, -\frac{2}{3\rho}\lambda_n(\nabla^2 f(\mathbf{x}_t + \boldsymbol{\Delta}_t^\star)) - \frac{2c_2}{3\sqrt{\rho}}\sqrt{\epsilon}\right\}.$$

We consider the case of large gradient and large Hessian in turn (one of which must hold since $\mathbf{x}_t + \boldsymbol{\Delta}_t^\star$ is not an $\epsilon$-second-order stationary point). There exist $c_1, c_2$ in the following so that we can obtain:

- If $\|\nabla f(\mathbf{x}_t + \boldsymbol{\Delta}_t^\star)\| > \epsilon$, then we have that

$$\|\boldsymbol{\Delta}_t^\star\| > \sqrt{\frac{1}{\rho(1 + \frac{c_2}{2})}\left(\|\nabla f(\mathbf{x}_t + \boldsymbol{\Delta}_t^\star)\| - (c_1 + \frac{c_2}{2})\epsilon\right)} \geq \sqrt{\frac{1 - c_1 - \frac{c_2}{2}}{1 + \frac{c_2}{2}}}\sqrt{\frac{\epsilon}{\rho}} > \frac{1}{2}\sqrt{\frac{\epsilon}{\rho}}. \tag{17}$$

- If $-\lambda_n(\nabla^2 f(\mathbf{x}_t + \boldsymbol{\Delta}_t^\star)) > \sqrt{\rho\epsilon}$, then we have that $\|\boldsymbol{\Delta}_t^\star\| > \frac{2}{3}\sqrt{\frac{\epsilon}{\rho}} - \frac{2c_2}{3}\sqrt{\frac{\epsilon}{\rho}} = \frac{2}{3}(1 - c_2)\sqrt{\frac{\epsilon}{\rho}} > \frac{1}{2}\sqrt{\frac{\epsilon}{\rho}}$.

We can similarly check the lower bounds directly stated are true. Choosing $c_1 = \frac{1}{200}$ and $c_2 = \frac{1}{200}$ will verify these inequalities for example. □

## B.3 Proof of Claim 1

Here we provide a proof of statement equivalent to Claim 1 in the full, non-stochastic setting with approximate model minimization. We focus on the case when the Cubic-Subsolver routine executes **Case 2**, since the result is vacuously true when the routine executes **Case 1**. Our first lemma will both help show sufficient descent and provide a stopping condition for Algorithm 1. For context, recall that when $\mathbf{x}_t + \boldsymbol{\Delta}_t^\star$ is not an $\epsilon$-second-order stationary point then $\|\boldsymbol{\Delta}_t^\star\| \geq \frac{1}{2}\sqrt{\frac{\epsilon}{\rho}}$ by Lemma 4.

**Lemma 5.** *If the routine Cubic-Subsolver uses* **Case 2**, *and if* $\|\boldsymbol{\Delta}_t^\star\| \geq \frac{1}{2}\sqrt{\frac{\epsilon}{\rho}}$, *then it will return a point* $\boldsymbol{\Delta}$ *satisfying* $m_t(\mathbf{x}_t + \boldsymbol{\Delta}_t) \leq m_t(\mathbf{x}_t) - \frac{1 - c_3}{12}\rho\|\boldsymbol{\Delta}_t^\star\|^3 \leq \frac{1 - c_3}{96}\sqrt{\frac{\epsilon^3}{\rho}}$.

*Proof.* In the case when $\|\boldsymbol{\Delta}_t^\star\| \geq \frac{1}{2}\sqrt{\frac{\epsilon}{\rho}}$, by the definition of the routine Cubic-Subsolver we can ensure that $m_t(\mathbf{x}_t + \boldsymbol{\Delta}_t) \leq m_t(\mathbf{x}_t + \boldsymbol{\Delta}_t^\star) + \frac{c_3}{12}\rho\|\boldsymbol{\Delta}_t^\star\|^3$ for arbitarily small $c_3$ using $\mathcal{T}(\epsilon)$ iterations. We can now combine the aforementioned display with Lemma 3 (recalling that $m_t(\mathbf{x}_t) = f(\mathbf{x}_t)$) to conclude that:

$$m_t(\mathbf{x}_t + \boldsymbol{\Delta}_t) \leq m_t(\mathbf{x}_t + \boldsymbol{\Delta}_t^\star) + \frac{c_3}{12}\rho\|\boldsymbol{\Delta}_t^\star\|^3$$
$$m_t(\mathbf{x}_t + \boldsymbol{\Delta}_t^\star) \leq m_t(\mathbf{x}_t) - \frac{\rho}{12}\|\boldsymbol{\Delta}_t^\star\|^3 \implies \tag{18}$$
$$m_t(\mathbf{x}_t + \boldsymbol{\Delta}_t) \leq m_t(\mathbf{x}_t) - (\frac{1 - c_3}{12})\rho\|\boldsymbol{\Delta}_t^\star\|^3 \leq m_t(\mathbf{x}_t) - \frac{(1 - c_3)}{96}\sqrt{\frac{\epsilon^3}{\rho}}. \tag{19}$$

for suitable choice of $c_3$ which can be made arbitrarily small. □

**Claim 1.** *Assume we are in the setting of Lemma 2 with sufficiently small constants $c_1, c_2$. If $\boldsymbol{\Delta}$ is the output of the routine Cubic-Subsolver when executing **Case 2** and if $\mathbf{x}_t + \boldsymbol{\Delta}_t^\star$ is not an $\epsilon$-second-order stationary point of $f$, then $m_t(\mathbf{x}_t + \boldsymbol{\Delta}_t) - m_t(\mathbf{x}_t) \leq -\frac{1-c_3}{96}\sqrt{\frac{\epsilon^3}{\rho}}$.*

*Proof.* This is an immediate consequence of Lemmas 4 and 5. $\qquad\square$

If we do not observe sufficient descent in the cubic submodel (which is not possible in **Case 1** by definition) then as a consequence of Claim 1 and Lemma 5 we can conclude that $\|\boldsymbol{\Delta}_t^\star\| \leq \frac{1}{2}\sqrt{\frac{\epsilon}{\rho}}$ and that $\mathbf{x}_t + \boldsymbol{\Delta}_t^\star$ *is* an $\epsilon$-second-order stationary point. However, we cannot compute $\boldsymbol{\Delta}_t^\star$ directly. So instead we use a final gradient descent loop in Algorithm 3, to ensure the final point returned in this scenario will be an $\epsilon$-second-order stationary point up to a rescaling.

**Lemma 6.** *Assume we are in the setting of Lemma 2 with sufficiently small constants $c_1, c_2$. If $\mathbf{x}_t + \boldsymbol{\Delta}_t^\star$ is an $\epsilon$-second-order stationary point of $f$, and $\|\boldsymbol{\Delta}_t^\star\| \leq \frac{1}{2}\sqrt{\frac{\epsilon}{\rho}}$, then Algorithm 3 will output a point $\boldsymbol{\Delta}$ such that $\mathbf{x}_{t+1} = \mathbf{x}_t + \boldsymbol{\Delta}$ is a $4\epsilon$-second-order stationary point of $f$.*

*Proof.* Since $\mathbf{x}_t + \boldsymbol{\Delta}_t^\star$ is an $\epsilon$-second order stationary point of $f$, by gradient smoothness and the concentration conditions we have that $\|\mathbf{g}_t\| \leq \|\nabla f(\mathbf{x}_t + \boldsymbol{\Delta}_t^\star)\| + \ell\|\boldsymbol{\Delta}_t^\star\| + \|\mathbf{g}_t - \nabla f(\mathbf{x}_t)\| \leq (1 + c_1)\epsilon + \frac{1}{2}\sqrt{\frac{\epsilon}{\rho}}\ell \leq (\frac{3}{2} + 1 + c_1)\frac{\ell^2}{\rho} \leq \frac{19}{16}\frac{\ell^2}{\rho}$ for sufficiently small $c_1$. Then we can verify the step-size choice $\eta = \frac{1}{20}\ell$ and initialization at $\boldsymbol{\Delta} = 0$ (in the centered coordinates) for the routine Cubic-FinalSubsolver verifies Assumptions A and B[3] in Carmon and Duchi [2016]. So, by Corollary 2.5 in Carmon and Duchi [2016]—which states the norms of the gradient descent iterates, $\|\boldsymbol{\Delta}\|$, are non-decreasing and satisfy $\|\boldsymbol{\Delta}\| \leq \|\boldsymbol{\Delta}_t^\star\|$—we have that $\|\boldsymbol{\Delta} - \boldsymbol{\Delta}_t^\star\| \leq 2\|\boldsymbol{\Delta}_t^\star\| \leq \sqrt{\frac{\epsilon}{\rho}}$.

We first show that $-\lambda_{\min}(\nabla^2 f(\mathbf{x}_{t+1})) \lesssim \sqrt{\rho\epsilon}$. Since $f$ is $\rho$-Hessian-Lipschitz we have that:

$$\nabla^2 f(\mathbf{x}_{t+1}) \succeq \nabla^2 f(\mathbf{x}_t + \boldsymbol{\Delta}_t^\star) - \rho 2\|\boldsymbol{\Delta}_t^\star\|I \succeq -2\sqrt{\rho\epsilon}I.$$

We now show that $\|\nabla f(\mathbf{x}_{t+1})\| \lesssim \epsilon$ and thus also small. Once again using that $f$ is $\rho$-Hessian-Lipschitz (Lemma 1 in Nesterov and Polyak [2006]) we have that:

$$\left\|\nabla f(\mathbf{x}_{t+1}) - \nabla f(\mathbf{x}_t) - \nabla^2 f(\mathbf{x}_t)\boldsymbol{\Delta}\right\| \leq \frac{\rho}{2}\|\boldsymbol{\Delta}\|^2 \leq \frac{\rho}{2}\|\boldsymbol{\Delta}_t^\star\|^2 \leq \frac{\epsilon}{8}.$$

Now, by the termination condition in Algorithm 3 we have that $\left\|\mathbf{g} + \mathbf{B}\boldsymbol{\Delta} + \frac{\rho}{2}\|\boldsymbol{\Delta}\|\boldsymbol{\Delta}\right\| < \frac{\epsilon}{2}$. So,

$$\|\mathbf{g} + \mathbf{B}\boldsymbol{\Delta}\| < \frac{\epsilon}{2} + \frac{\rho}{2}\|\boldsymbol{\Delta}\|^2 \leq \frac{5}{8}\epsilon.$$

Using gradient/Hessian concentration with the previous displays we also obtain that:

$$\|\nabla f(\mathbf{x}_{t+1})\| - \|\mathbf{g} - \nabla f(\mathbf{x}_t)\| - \left\|(\mathbf{B} - \nabla^2 f(\mathbf{x}_t))\boldsymbol{\Delta}\right\| - \|\mathbf{g} + \mathbf{B}\boldsymbol{\Delta}\| \leq \left\|\nabla f(\mathbf{x}_{t+1}) - \nabla f(\mathbf{x}_t) - \nabla^2 f(\mathbf{x}_t)\boldsymbol{\Delta}\right\|$$

$$\implies \|\nabla f(\mathbf{x}_{t+1})\| \leq \left(c_1 + \frac{c_2}{2} + \frac{5}{8} + \frac{1}{8}\right)\epsilon \leq \epsilon,$$

for sufficiently small $c_1$ and $c_2$.

Let us now bound the iteration complexity of this step. From our previous argument we have that $\|\mathbf{g}_t\| \leq (1 + c_1)\epsilon + \frac{\ell}{2\sqrt{\rho}}\sqrt{\epsilon}$. Similarly, the concentration conditions imply $\|\mathbf{B}_t\boldsymbol{\Delta}_t^\star\| \leq (\ell + c_2\sqrt{\rho\epsilon})\|\boldsymbol{\Delta}_t^\star\|$. Thus we have that $m_t(\mathbf{x}_t) - m_t(\mathbf{x}_t + \boldsymbol{\Delta}_t^\star) = ((1 + c_1)\epsilon + \frac{\ell}{2\sqrt{\rho}}\sqrt{\epsilon})\|\boldsymbol{\Delta}_t^\star\| + \frac{1}{2}(\ell + c_2\sqrt{\rho\epsilon})\|\boldsymbol{\Delta}_t^\star\|^2 + \frac{\rho}{6}\|\boldsymbol{\Delta}_t^\star\|^3 \leq \frac{3\ell}{\rho}\epsilon + \left(\frac{1+c_1+4c_2}{8} + \frac{1}{48}\right)\sqrt{\frac{\epsilon^3}{\rho}} \leq \mathcal{O}(1) \cdot \frac{\epsilon\ell}{\rho}$ since $c_1, c_2$ are numerical constants that can be made arbitrarily small.

So by the standard analysis of gradient descent for smooth functions, see Nesterov [2013] for example, we have that Algorithm 3 will terminate in at most $\leq \lceil \frac{m_t(\mathbf{x}_t) - m_t(\mathbf{x}_t + \boldsymbol{\Delta}_t^\star)}{\eta(\epsilon/2)^2} \rceil \leq \mathcal{O}(1) \cdot (\frac{\ell^2}{\rho\epsilon})$ iterations. This will take at most $\tilde{\mathcal{O}}(\max(\frac{M_1}{\sqrt{\rho\epsilon}}, \frac{\sigma_2^2}{\epsilon}) \cdot \frac{\ell^2}{\rho\epsilon})$ Hessian-vector products and $\tilde{\mathcal{O}}(\max(\frac{M_1}{\epsilon}, \frac{\sigma_1^2}{\epsilon^2}))$ gradient evaluations which will be subleading in the overall complexity. $\qquad\square$

## B.4 Proof of Claim 2

We now prove our main descent lemma equivalent to **Claim 2**—this will show that if the cubic submodel has a large decrease, then the underlying true function must also have large decrease. As before we focus on the case when the Cubic-Subsolver routine executes **Case 2** since the result is vacuously true in **Case 1**.

**Claim 2.** *Assume we are in the setting of Lemma 2 with sufficiently small constants $c_1, c_2$. If the Cubic-Subsolver routine uses **Case 2**, and if $m_t(\mathbf{x}_t + \mathbf{\Delta}_t) - m_t(\mathbf{x}_t) \leq -(\frac{1-c_3}{96})\sqrt{\frac{\epsilon^3}{\rho}}$, then*

$$f(\mathbf{x}_t + \mathbf{\Delta}_t) - f(\mathbf{x}_t) \leq -\left(\frac{1-c_3-c_5}{96}\right)\sqrt{\frac{\epsilon^3}{\rho}}.$$

*Proof.* Using that $f$ is $\rho$-Hessian Lipschitz (and hence admits a cubic majorizer by Lemma 1 in Nesterov and Polyak [2006] for example) as well as the concentration conditions we have that:

$$f(\mathbf{x}_t + \mathbf{\Delta}_t) \leq f(\mathbf{x}_t) + \nabla f(\mathbf{x}_t)^\top \mathbf{\Delta}_t + \frac{1}{2}\mathbf{\Delta}_t^\top \nabla^2 f(\mathbf{x}_t)\mathbf{\Delta}_t + \frac{\rho}{6}\|\mathbf{\Delta}_t\|_2^3 \implies$$

$$f(\mathbf{x}_t + \mathbf{\Delta}_t) - f(\mathbf{x}_t) \leq m_t(\mathbf{x}_t + \mathbf{\Delta}_t) - m_t(\mathbf{x}_t) + (\nabla f(\mathbf{x}_t) - \mathbf{g}_t)^\top \mathbf{\Delta}_t + \frac{1}{2}\mathbf{\Delta}_t^\top (\mathbf{B}_t - \nabla^2 f(\mathbf{x}_t))\mathbf{\Delta}_t$$

$$\leq m_t(\mathbf{x}_t + \mathbf{\Delta}_t) - m_t(\mathbf{x}_t) + c_1\epsilon\|\mathbf{\Delta}_t\| + \frac{c_2}{2}\sqrt{\rho\epsilon}\|\mathbf{\Delta}_t\|^2$$

$$\leq m_t(\mathbf{x}_t + \mathbf{\Delta}_t) - m_t(\mathbf{x}_t) + c_1\epsilon\left(\|\mathbf{\Delta}_t^\star\| + c_4\sqrt{\frac{\epsilon}{\rho}}\right) + \frac{c_2}{2}\sqrt{\rho\epsilon}\left(\|\mathbf{\Delta}_t^\star\| + c_4\sqrt{\frac{\epsilon}{\rho}}\right)^2$$

$$\leq m_t(\mathbf{x}_t + \mathbf{\Delta}_t) - m_t(\mathbf{x}_t) + (c_1 + c_2c_4)\epsilon\|\mathbf{\Delta}_t^\star\| + \frac{c_2c_4^2}{2}\sqrt{\rho\epsilon}\|\mathbf{\Delta}_t^\star\|^2 + (c_1 + \frac{c_2c_4}{2})c_4\sqrt{\frac{\epsilon^3}{\rho}}, \quad (20)$$

since by the definition the Cubic-Subsolver routine, when we use **Case 2** we have that $\|\mathbf{\Delta}_t\| \leq \|\mathbf{\Delta}_t^\star\| + c_4\sqrt{\frac{\epsilon}{\rho}}$. We now consider two different situations – when $\|\mathbf{\Delta}_t^\star\| \geq \frac{1}{2}\sqrt{\frac{\epsilon}{\rho}}$ and when $\|\mathbf{\Delta}_t^\star\| \leq \frac{1}{2}\sqrt{\frac{\epsilon}{\rho}}$.

First, if $\|\mathbf{\Delta}_t^\star\| \geq \frac{1}{2}\sqrt{\frac{\epsilon}{\rho}}$ then by Lemma 5 we may assume the stronger guarantee that $m_t(\mathbf{x}_t + \mathbf{\Delta}_t) - m_t(\mathbf{x}_t) \leq -(\frac{1-c_3}{12})\rho\|\mathbf{\Delta}_t^\star\|^3$. So by considering the above display in Equation (20) we can conclude that:

$$f(\mathbf{x}_t + \mathbf{\Delta}_t) - f(\mathbf{x}_t) \leq m_t(\mathbf{x}_t + \mathbf{\Delta}_t) - m_t(\mathbf{x}_t) + (c_1 + c_2c_4)\epsilon\|\mathbf{\Delta}_t^\star\| + \frac{c_2c_4^2}{2}\sqrt{\rho\epsilon}\|\mathbf{\Delta}_t^\star\|^2 + (c_1 + \frac{c_2c_4}{2})c_4\sqrt{\frac{\epsilon^3}{\rho}}$$

$$\leq -\left(\frac{1 - c_3 - 48(c_1 + c_2c_4) - 12c_2c_4^2}{12}\right)\rho\|\mathbf{\Delta}_t^\star\|^3 + \left(c_1 + \frac{c_2c_4}{2}\right)c_4\sqrt{\frac{\epsilon^3}{\rho}}$$

$$\leq -\left(\frac{1 - c_3 - 48c_1 - 48c_2c_4 - 96c_1c_4 - 60c_2c_4^2}{96}\right)\sqrt{\frac{\epsilon^3}{\rho}},$$

since the numerical constants $c_1, c_2, c_3$ can be made arbitrarily small.

Now, if $\|\mathbf{\Delta}_t^\star\| \leq \frac{1}{2}\sqrt{\frac{\epsilon}{\rho}}$, we directly use the assumption that $m_t(\mathbf{x}_t + \mathbf{\Delta}_t) - m_t(\mathbf{x}_t) \leq -(\frac{1-c_3}{96})\sqrt{\frac{\epsilon^3}{\rho}}$. Combining with the display in in Equation (20) we can conclude that:

$$f(\mathbf{x}_t + \mathbf{\Delta}_t) - f(\mathbf{x}_t) \leq m_t(\mathbf{x}_t + \mathbf{\Delta}_t) - m_t(\mathbf{x}_t) + (c_1 + c_2c_4)\epsilon\|\mathbf{\Delta}_t^\star\| + \frac{c_2c_4^2}{2}\sqrt{\rho\epsilon}\|\mathbf{\Delta}_t^\star\|^2 + (c_1 + \frac{c_2c_4}{2})c_4\sqrt{\frac{\epsilon^3}{\rho}}$$

$$\leq -\left(\frac{1 - c_3}{96}\sqrt{\frac{\epsilon^3}{\rho}}\right) + \left((c_1 + c_2c_4)\epsilon \cdot \frac{1}{2}\sqrt{\frac{\epsilon}{\rho}} + \frac{c_2c_4^2}{2}\sqrt{\rho\epsilon}\cdot\frac{1}{4}\frac{\epsilon}{\rho} + (c_1 + \frac{c_2c_4}{2})c_4\sqrt{\frac{\epsilon^3}{\rho}}\right)$$

$$\leq -\left(\frac{1 - c_3 - 48c_1 - 48c_2c_4 - 96c_1c_4 - 60c_2c_4^2}{96}\right)\sqrt{\frac{\epsilon^3}{\rho}},$$

since the numerical constants $c_1, c_2, c_3$ can be made arbitrarily small. Indeed, recall that $c_1$ is the gradient concentration constant, $c_2$ is the Hessian-vector product concentration constant, and $c_3$ is the tolerance of the Cubic-Subsolver routine when using **Case 2**. Thus, in both situations, we have that:

$$f(\mathbf{x}_t + \boldsymbol{\Delta}_t) - f(\mathbf{x}_t) \le \frac{1 - c_3 - c_5}{96} \sqrt{\frac{\epsilon^3}{\rho}}, \tag{21}$$

denoting $c_5 = 48c_1 - 48c_2c_4 - 96c_1c_4 - 60c_2c_4^2$ for notational convenience (which can also be made arbitrarily small for sufficiently small $c_1, c_2$). $\quad\square$

### B.5 Proof of Theorem 1

We now prove the correctness of Algorithm 1. We assume, as usual, the underlying function $f(x)$ possesses a lower bound $f^*$.

**Theorem 1.** *There exists an absolute constant $c$ such that if $f(\mathbf{x})$ satisfies Assumptions 1, 2, CubicSubsolver satisfies Condition 1 with $c$, $n_1 \ge \max(\frac{M_1}{c\epsilon}, \frac{\sigma_1^2}{c^2\epsilon^2}) \log\left(\frac{d\sqrt{\rho}\Delta_f}{\epsilon^{1.5}\delta c}\right)$, and $n_2 \ge \max(\frac{M_2}{c\sqrt{\rho\epsilon}}, \frac{\sigma_2^2}{c^2\rho\epsilon}) \log\left(\frac{d\sqrt{\rho}\Delta_f}{\epsilon^{1.5}\delta c}\right)$, then for all $\delta > 0$ and $\Delta_f \ge f(\mathbf{x}_0) - f^*$, Algorithm 1 will output an $\epsilon$-second-order stationary point of $f$ with probability at least $1 - \delta$ within*

$$\tilde{\mathcal{O}}\left(\frac{\sqrt{\rho}\Delta_f}{\epsilon^{1.5}}\left(\max\left(\frac{M_1}{\epsilon}, \frac{\sigma_1^2}{\epsilon^2}\right) + \max\left(\frac{M_2}{\sqrt{\rho\epsilon}}, \frac{\sigma_2^2}{\rho\epsilon}\right) \cdot \mathcal{T}(\epsilon)\right)\right) \tag{5}$$

*total stochastic gradient and Hessian-vector product evaluations.*

*Proof.* For notational convenience let **Case 1** of the routine Cubic Subsolver satisfy:

$$\max\{f(\mathbf{x}_t + \boldsymbol{\Delta}_t) - f(\mathbf{x}_t), m_t(\mathbf{x}_t + \boldsymbol{\Delta}_t) - m_t(\mathbf{x}_t)\} \le -K_1\sqrt{\frac{\epsilon^3}{\rho}}.$$

and use $K_2 = \frac{1-c_3}{96}$ to denote the descent constant of the cubic submodel in the assumption of Claim 2. Further, let $K_{\text{prog}} = \min\{\frac{1-c_3-c_5}{96}, K_1\}$ which we will use as the progress constant corresponding to descent in the underlying function $f$. Without loss of generality, we assume that $-K_1 \le -K_2$ for convenience in the proof. If $-K_1 \ge -K_2$, we can simply rescale the descent constant corresponding to **Case 2** for the cubic submodel, $\frac{1-c_3}{96}$, to be equal to $-K_1$, which will require shrinking $c_1, c_2$ proportionally to ensure that the rescaled version of the function descent constant, $\frac{1-c_3-c_5}{96}$, is positive.

Now, we choose $c_1, c_2, c_3$ so that $K_2 > 0$, $K_{\text{prog}} > 0$, and Lemma 4 holds in the aforementioned form. For the correctness of Algorithm 1 we choose the numerical constant in Line 7 as $K_2$ – so the "if statement" checks the condition $\Delta m = m_t(\mathbf{x}_{t+1}) - m_t(\mathbf{x}_t) \ge -K_2\sqrt{\frac{\epsilon'^3}{\rho}}$. Here we use a rescaled $\epsilon' = \frac{1}{4}\epsilon$ for the duration of the proof.

At each iteration the event that the setting of Lemma 2 hold has probability greater then $1 - 2\delta'$. Conditioned on this event let the routine Cubic-Subsolver have a further probability of at most $\delta'$ of failure. We now proceed with our analysis deterministically conditioned on the event $E$ – that at each iteration the concentration conditions hold and the routine Cubic-Subsolver succeeds – which has probability greater then $1 - 3\delta'T_{\text{outer}} \ge 1 - \delta$ by a union bound for $\delta' = \frac{\delta}{3T_{\text{outer}}}$.

Let us now bound the iteration complexity of Algorithm 1 as $T_{\text{outer}}$. We cannot have the "if statement" in Line 7 fail indefinitely. At a given iteration, if the routine Cubic-Subsolver outputs a point $\boldsymbol{\Delta}$ that satisfies

$$m_t(\mathbf{x}_t + \boldsymbol{\Delta}_t) - m_t(\mathbf{x}_t) \le -K_2\sqrt{\frac{\epsilon'^3}{\rho}}$$

then by Claim 2 and the definition of **Case 1** of the Cubic-Subsolver we also have that:

$$f(\mathbf{x}_t + \boldsymbol{\Delta}_t) - f(\mathbf{x}_t) \leq -K_{\text{prog}}\sqrt{\frac{\epsilon'^3}{\rho}}.$$

Note if the Cubic-Subsolver uses **Case 1** in this iteration then we will vacuously achieve descent in both the underlying function $f$, and descent in the cubic submodel greater $-K_1\sqrt{\frac{\epsilon'^3}{\rho}}$. Since $-K_1\sqrt{\frac{\epsilon'^3}{\rho}} \leq -K_2\sqrt{\frac{\epsilon'^3}{\rho}}$ by assumption, the algorithm will not terminate early at this iteration. Since the function $f$ is bounded below by $f^*$, the event $m_t(\mathbf{x}_t + \boldsymbol{\Delta}_t) - m_t(\mathbf{x}_t) \leq -K_2\sqrt{\frac{\epsilon'^3}{\rho}}$ which implies $f(\mathbf{x}_{t+1}) - f(\mathbf{x}_t) \leq -K_{\text{prog}}\sqrt{\frac{\epsilon'^3}{\rho}}$ can happen at most $T_{\text{outer}} = \lceil \frac{\sqrt{\rho}(f(x_0)-f^*)}{K_{\text{prog}}\epsilon'^{1.5}} \rceil$ times.

Thus in the $T_{\text{outer}}$ iterations of Algorithm 1 it must be the case that there is at least one iteration $T$, for which

$$m_T(\mathbf{x}_T + \boldsymbol{\Delta}_T) - m_T(\mathbf{x}_T) \geq -K_2\sqrt{\frac{\epsilon'^3}{\rho}}.$$

By the definition of the Cubic-Subsolver procedure and assumption that $-K_1 \leq -K_2$, it must be the case at iteration $T$ the routine Cubic-Subsolver used **Case 2**. Now by appealing to Claim 1 and Lemma 5 we must have that $\|\boldsymbol{\Delta}_T^\star\| \leq \frac{1}{2}\sqrt{\frac{\epsilon'}{\rho}}$ and that $\mathbf{x}_T + \boldsymbol{\Delta}_T^\star$ is an $\epsilon'$-second-order stationary point of $f$. As we can see in Line 7 of Algorithm 1, at iteration $T$ the "if statement" will be true. Hence Algorithm 1 will run the final gradient descent loop (Algorithm 3) at iteration $T$, return the final point and proceed to exit via the break statement. Since the hypotheses of Lemma 6 are satisfied[4] at iteration $T$, Algorithm 3 will return a final point that is an $\epsilon$-second-order stationary point of $f$ as desired. We can verify the global constant $c = \min\{\frac{K_{\text{prog}}}{8}, c_1, c_2\}$ satisfies the conditions of the theorem.

**Remark 3.** We can also now do a careful count of the complexity of Algorithm 1. First, note at each outer iteration of Algorithm 1 we require $n_1 \geq \max\left(\frac{M_1}{c_1\epsilon}, \frac{\sigma_1^2}{c_1^2\epsilon^2}\right)\frac{8}{3}\log\frac{2d}{\delta'}$ samples to approximate the gradient and and $n_2 \geq \max(\frac{M_2}{c_2\sqrt{\rho\epsilon}}, \frac{\sigma_2^2}{c_2^2\rho\epsilon})\frac{8}{3}\log\frac{2d}{\delta'}$ to approximate the Hessian. The union bound stipulates we should take $\delta'(\epsilon) = \frac{\delta}{3T_{\text{outer}}}$ to control the total failure probability of Algorithm 1. Then as we can see in the Proof of Theorem 1, Algorithm 1 will terminate in at most

$$T_{\text{outer}} = \lceil \frac{8K_{\text{prog}}\sqrt{\rho}(f(x_0)-f^*)}{\epsilon^{3/2}} \rceil \tag{22}$$

iterations. The inner iteration complexity of the Cubic-Subsolver routine is $\mathcal{T}(\epsilon)$. The routine only requires computing the gradient vector once, but recomputes Hessian-vector products at each iteration.

So the gradient complexity becomes

$$T_{\text{G}} \lesssim \mathcal{T}(\epsilon) \times \frac{\sqrt{\rho}(f(x_0)-f^*)}{\epsilon^{1.5}} \times \max\left(\frac{M_1}{c_1\epsilon}, \frac{\sigma_1^2}{c_1^2\epsilon^2}\right)\frac{8}{3}\log\frac{2d}{\delta'}$$

$$\sim \tilde{\mathcal{O}}\left(\frac{\sqrt{\rho}\sigma_1^2(f(x_0)-f^*)}{\epsilon^{3.5}}\right) \text{ for } \epsilon \leq \frac{\sigma_1^2}{c_1 M_1}.$$

Note that $\tilde{O}$ hides logarithmic factors since $\delta'(\epsilon) = \frac{\delta}{3T_{\text{outer}}}$.

The total complexity of Hessian-vector product evaluations is:

$$T_{\text{HV}} \lesssim \frac{K_{\text{prog}}\sqrt{\rho}(f(x_0)-f^*)}{\epsilon^{1.5}} \times \max(\frac{M_2}{c_2\sqrt{\rho\epsilon}}, \frac{\sigma_2^2}{c_2^2\rho\epsilon})\frac{8}{3}\log\frac{2d}{\delta'}$$

$$\sim \tilde{\mathcal{O}}\left(\mathcal{T}(\epsilon)\frac{\sigma_2^2(f(x_0)-f^*)}{\sqrt{\rho}\epsilon^3}\right) \text{ for } \epsilon \leq \frac{\sigma_2^4}{c_2^2 M_2^2\rho}.$$

Finally, recall the proof of Lemma 6 which shows total complexity of the final gradient descent loop, in Algorithm 3, will be subleading in overall gradient and Hessian-vector product complexity. As before, we can verify the global constant $c = \min\{\frac{K_{\text{prog}}}{8}, c_1, c_2\}$ satisfies the conditions of the Theorem 1.

<div align="right">□</div>

## C  Gradient Descent as a Cubic Subsolver

Here we provide the proofs of Lemma 1 and Corollary 1. In particular, we show Algorithm 2 is a Cubic-Subsolver routine satisfying Condition 1. We break the analysis into two cases showing that whenever $\|\mathbf{g}\| \geq \frac{\ell^2}{\rho}$ taking a Cauchy step satisfies the conditions of a **Case 1** procedure, and when $\|\mathbf{g}\| \leq \frac{\ell^2}{\rho}$, running gradient descent on the cubic submodel satisfies the conditions of a **Case 2** procedure. Showing the latter relies on Theorem 3.2 from Carmon and Duchi [2016].

### C.1  Cauchy Step in Algorithm 2

First, we argue that when the stochastic gradients are large – $\|\mathbf{g}_t\| \geq \frac{\ell^2}{\rho}$ – the Cauchy step achieves sufficient descent in both $f$ and $\tilde{m}$. Thus the Cauchy step will satisfy Equation 1 of Condition 1, so it satisfies the conditions of a **Case 1** procedure. As before, $\tilde{m}(\mathbf{\Delta}) = \mathbf{\Delta}^\top \mathbf{g} + \frac{1}{2}\mathbf{\Delta}^\top \mathbf{B}[\mathbf{\Delta}] + \frac{\rho}{6}\|\mathbf{\Delta}\|^3$, refers to stochastic cubic submodel in centered coordinates.

First, recall several useful results: the *Cauchy radius* is the magnitude of the global minimizer in the subspace spanned by $\mathbf{g}$ (which is analytically tractable for this cubic submodel). Namely $R_c = \operatorname{argmin}_{\eta \in \mathbb{R}^d} \tilde{m}\left(-\eta \frac{\mathbf{g}}{\|\mathbf{g}\|}\right)$ and a short computation shows that

$$R_c = -\frac{\mathbf{g}^\top \mathbf{B} \mathbf{g}}{\rho \|\mathbf{g}\|^2} + \sqrt{\left(\frac{\mathbf{g}^\top \mathbf{B} \mathbf{g}}{\rho \|\mathbf{g}\|^2}\right)^2 + \frac{2\|\mathbf{g}\|}{\rho}}$$

and that

$$\tilde{m}\left(-R_c \frac{\mathbf{g}}{\|\mathbf{g}\|}\right) = -\frac{1}{2}\|\mathbf{g}\|R_c - \frac{\rho}{12}R_c^3.$$

**Lemma 7.** *Assume we are in the setting of Lemma 2 with sufficiently small constants $c_1, c_2$. If $\|\mathbf{g}\| \geq \frac{\ell^2}{\rho}$, the Cauchy step defined by taking $\mathbf{\Delta} = -R_c \frac{\mathbf{g}}{\|\mathbf{g}\|}$ will satisfy $\tilde{m}(\mathbf{\Delta}) \leq -K_1 \sqrt{\frac{\epsilon^3}{\rho}}$ and $f(\mathbf{x}_t + \mathbf{\Delta}) - f(\mathbf{x}_t) \leq -K_1 \sqrt{\frac{\epsilon^3}{\rho}}$ for $K_1 = -\frac{7}{20}$. Thus, if Algorithm 2 takes a Cauchy step (when $\|\mathbf{g}\| \geq \frac{\ell^2}{\rho}$) it will satisfy the conditions of an **Case 1** procedure.*

*Proof.* We first lower bound the Cauchy radius

$$R_c = -\frac{\mathbf{g}^\top \mathbf{B} \mathbf{g}}{\rho \|\mathbf{g}\|^2} + \sqrt{\left(\frac{\mathbf{g}^\top \mathbf{B} \mathbf{g}}{\rho \|\mathbf{g}\|^2}\right)^2 + \frac{2\|\mathbf{g}\|}{\rho}} \geq \frac{1}{\rho}\left(-\frac{\mathbf{g}^\top \mathbf{B} \mathbf{g}}{\|\mathbf{g}\|^2} + \sqrt{\left(\frac{\mathbf{g}^\top \mathbf{B} \mathbf{g}}{\|\mathbf{g}\|^2}\right)^2 + 2\ell^2}\right).$$

Note the function $-x + \sqrt{x^2 + 2}$ is decreasing over its support. Additionally, $\|\nabla^2 f(0)\| \leq \ell$ and $\|(\mathbf{B} - \nabla^2 f(0))\mathbf{g}\| \leq c_2\sqrt{\rho\epsilon}\|\mathbf{g}\| \leq c_2\ell\|\mathbf{g}\|$ imply that $\frac{\mathbf{g}^\top \mathbf{B} \mathbf{g}}{\|\mathbf{g}\|^2} \leq (1 + c_2)\ell$. So, combining with the previous display we obtain:

$$R_c \geq \frac{\ell}{\rho}\left(-(1 + c_2) + \sqrt{(1 + c_2)^2 + 2}\right) \geq \frac{7}{10}\frac{\ell}{\rho}, \tag{23}$$

for sufficiently small $c_2$. So,

$$\tilde{m}\left(-R_c \frac{\mathbf{g}}{\|\mathbf{g}\|}\right) = -\frac{1}{2}\|\mathbf{g}\|R_c - \frac{\rho}{12}R_c^3 \leq -\frac{7}{20}\frac{\ell^3}{\rho^2} \leq -\frac{7}{20}\sqrt{\frac{\epsilon^3}{\rho}},$$

<div align="center">20</div>

for sufficiently small $c_2$.

Now, at iteration $t$, using the $\rho$-Hessian Lipschitz condition and concentration conditions we obtain:

$$f\left(\mathbf{x}_t - R_c\frac{\mathbf{g}}{\|\mathbf{g}\|}\right) - f(\mathbf{x}_t) \leq \tilde{m}\left(-R_c\frac{\mathbf{g}}{\|\mathbf{g}\|}\right) + c_1 R_c \epsilon + \frac{1}{2}R_c^2 c_2\sqrt{\rho\epsilon}$$

$$\leq -\frac{1}{2}\|\mathbf{g}\|R_c - \frac{\rho}{12}R_c^3 + c_1 R_c \epsilon + \frac{1}{2}R_c^2 c_2\sqrt{\rho\epsilon}$$

$$\leq -\frac{1}{2}\|\mathbf{g}\|R_c - \frac{\ell^3}{\rho^2}\cdot\frac{\rho}{\ell}R_c\left(\frac{1}{12}(\frac{\rho}{\ell}R_c)^2 - c_1 - \frac{c_2}{2}(\frac{\rho}{\ell}R_c)\right).$$

For sufficiently small $c_1, c_2$ the quadratic $\frac{1}{12}x^2 - c_1 - \frac{c_2}{2}x$ is minimized at $x^* = \frac{7}{10}$ when restricted to $x \geq \frac{7}{10}$. Since $\frac{\rho}{\ell}R_c \geq \frac{7}{10}$ for sufficiently small $c_1, c_2$ as in Equation (23), we can also conclude that

$$f\left(\mathbf{x}_t - R_c\frac{\mathbf{g}}{\|\mathbf{g}\|}\right) - f(\mathbf{x}_t) \leq -\frac{7}{20}\frac{\ell^3}{\rho^2} \leq -\frac{7}{20}\sqrt{\frac{\epsilon^3}{\rho}}. \tag{24}$$

This establishes sufficient decrease with respect to the true function $f$. We can verify choosing $c_1 = c_2 = 1/200$ satisfies all the inequalities in this section. Lastly, note the complexity of computing the Cauchy step is in fact $\mathcal{O}(1)$. $\qquad\square$

## C.2 Gradient Descent Loop in Algorithm 2

We now establish complexity of sufficient descent when we are using the gradient descent subsolver from Carmon and Duchi [2016] to minimize the stochastic cubic submodel in Algorithm 2 in the regime $\|\mathbf{g}\| \leq \frac{\ell^2}{\rho}$. Carmon and Duchi [2016] consider the gradient descent scheme[5]:

$$\boldsymbol{\Delta}_{t+1} = \boldsymbol{\Delta}_t - \eta\nabla\tilde{m}(\boldsymbol{\Delta}_t) \tag{25}$$

for $\tilde{m}(\boldsymbol{\Delta}) = \mathbf{g}^\top\boldsymbol{\Delta} + \frac{1}{2}\boldsymbol{\Delta}^\top\mathbf{B}\boldsymbol{\Delta} + \frac{\rho}{6}\|\boldsymbol{\Delta}^3\|$. Here we let the index $t$ range over the iterates of the gradient descent loop for a fixed cubic submodel. Defining $\beta = \|\mathbf{B}\|$ and $R = \frac{\beta}{\rho} + \sqrt{(\frac{\beta}{\rho})^2 + \frac{2\|\mathbf{g}\|}{\rho}}$ they make the following assumptions to show convergence:

- **Assumption A**: The step size for the gradient descent scheme satisfies

$$0 < \eta < \frac{1}{4(\beta + \frac{\rho}{2}R)}. \tag{26}$$

- **Assumption B**: The initialization for the gradient descent scheme $\boldsymbol{\Delta}$, satisfies $\boldsymbol{\Delta} = -r\frac{\mathbf{g}}{\|\mathbf{g}\|}$, with $0 \leq r \leq R_c$.

Then, Theorem 3.2 in Carmon and Duchi [2016] (restated here for convenience) gives that:

**Theorem 2** (Carmon and Duchi [2016]). *Let $q$ be uniformly distributed on the the unit sphere in $\mathbb{R}^d$. Then, the iterates $\boldsymbol{\Delta}_t$ generated by the gradient descent scheme in Equation (25) satisfying Assumptions A and B obtained by replacing $\mathbf{g} \to \mathbf{g} + \sigma q$ for $\sigma = \frac{\rho\epsilon'}{\beta + \rho\|\boldsymbol{\Delta}^\star\|} \cdot \frac{\bar{\sigma}}{12}$ with $\bar{\sigma} \leq 1$, satisfy $\tilde{m}(\boldsymbol{\Delta}_t) \leq \tilde{m}(\boldsymbol{\Delta}^\star) + (1 + \bar{\sigma})\epsilon'$ for all:*

$$t \geq \mathcal{T}(\epsilon') = \frac{1 + \bar{\sigma}}{\eta}\min\left\{\frac{1}{\frac{\rho}{2}\|\boldsymbol{\Delta}^\star\| - \gamma}, \frac{10\|\boldsymbol{\Delta}^\star\|^2}{\epsilon'}\right\}\times$$

$$\left[6\log\left(1 + \mathbb{I}_{\{\gamma>0\}}\frac{3\sqrt{d}}{\bar{\sigma}\delta'}\right) + 14\log\left(\frac{(\beta + \rho\|\boldsymbol{\Delta}^\star\|)\|\boldsymbol{\Delta}^\star\|^2}{\epsilon'}\right)\right], \tag{27}$$

*with probability $1 - \delta'$.*

Note that the iterates $\boldsymbol{\Delta}_t$ are iterates generated from the solving the *perturbed* cubic subproblem with $\mathbf{g} \to \mathbf{g} + \sigma q$—not from solving the original cubic subproblem. This step is necessary to avoid the "hard case" of the non-convex quadratic problems. See Carmon and Duchi [2016] for more details.

To apply this bound we will never have access to $\|\boldsymbol{\Delta}^\star\|$ apriori. However, in the present we need only to use this lemma to conclude sufficient descent when $\boldsymbol{\Delta}^\star$ is not an $\epsilon$-second-order stationary point and hence when $\|\boldsymbol{\Delta}\|^\star \geq \frac{1}{2}\sqrt{\frac{\epsilon}{\rho}}$.

**Lemma 8.** *Assume we are in the setting of Lemma 2 with sufficiently small constants $c_1, c_2$. Further, assume that $\|\mathbf{g}\| \leq \frac{\ell^2}{\rho}$, and that we choose $\eta = \mathcal{O}(\frac{1}{\ell})$ in the gradient descent iterate scheme in Equation (25) with perturbed gradient $\tilde{\mathbf{g}} = \mathbf{g} + \sigma \mathbf{q}$ for $\sigma = \mathcal{O}(\frac{\sqrt{\epsilon\rho}}{\ell})$, and $q \sim Unif(S^{d-1})$. If $\|\boldsymbol{\Delta}^\star\| \geq \frac{1}{2}\sqrt{\frac{\epsilon}{\rho}}$, then the gradient descent iterate $\boldsymbol{\Delta}_{\mathcal{T}(\epsilon)}$ for $\mathcal{T}(\epsilon) = \tilde{\mathcal{O}}(\frac{\ell}{\sqrt{\rho\epsilon}})$ will satisfy $\tilde{m}(\boldsymbol{\Delta}) \leq \tilde{m}(\boldsymbol{\Delta}^\star) + \frac{c_3}{12}\|\boldsymbol{\Delta}^\star\|^3$. Further, the iterate $\boldsymbol{\Delta}_{\mathcal{T}(\epsilon)}$ will always satisfy $\|\boldsymbol{\Delta}_{\mathcal{T}(\epsilon)}\| \leq \|\boldsymbol{\Delta}^\star\| + c_4\sqrt{\frac{\epsilon}{\rho}}$.*

*Proof.* To show the first statement we simply invoke Theorem 3.2 of Carmon and Duchi [2016] and check that its assumptions are satisfied. We assume that $\|\boldsymbol{\Delta}^\star\| \geq \frac{1}{2}\sqrt{\frac{\epsilon}{\rho}} \equiv L$, $\|\mathbf{g}\| \leq \frac{\ell^2}{\rho}$ in the present situation. We now set $\epsilon' = c_3 \frac{\rho}{24}\|\boldsymbol{\Delta}^\star\|^3$ where we assume that $c_3$ and will be small. Then we take:

$$\sigma = c_3 \frac{\rho^2 L^3}{24 \cdot 12(2\ell + \rho L)} = \frac{\rho L^3 \epsilon'}{12\|\boldsymbol{\Delta}^\star\|^3(2\ell + \rho L)} = \frac{\rho \epsilon'}{\beta + \rho\|\boldsymbol{\Delta}^\star\|} \cdot \frac{1}{12} \underbrace{\frac{\beta + \rho\|\boldsymbol{\Delta}^\star\|}{2\ell + \rho L} \frac{L^3}{\|\boldsymbol{\Delta}^\star\|^3}}_{\bar{\sigma}}.$$

By concentration we have that $(1-c_2)\ell \leq \beta \leq (1+c_2)\ell$. So we can easily check that $\frac{1}{3}\frac{L^3}{\|\boldsymbol{\Delta}^\star\|^3} \leq \bar{\sigma} \leq 1$ for sufficiently small $c_2$. We can further show that:

$$\sigma = c_3 \frac{\rho^2 L^3}{24 \cdot 12(2\ell + \rho L)} \leq c_3 \frac{\rho^2 L^3}{24 \cdot 12\rho L} \leq c_3 \frac{\rho^2 L^3}{24 \cdot 12\rho L} \leq c_3 \frac{\rho L^2}{288} \leq c_3 \frac{\epsilon}{2304}. \tag{28}$$

Thus $R = \frac{\beta}{\rho} + \sqrt{(\frac{\beta}{\rho})^2 + \frac{2\|\mathbf{g}\|}{\rho}} \leq 2\frac{\ell}{\rho}$ for sufficiently small $c_1$, $c_2$, and $c_4$. Accordingly, $\frac{1}{4(\beta + \frac{\rho}{2}R)} \geq \frac{1}{20\ell}$. We can similarly check this step size choice suffices for Lemma 6. Let us then choose $\eta = \frac{1}{20\ell}$ to satisfy **Assumption A** and $r = 0$ in accordance with **Assumption B** from Carmon and Duchi [2016].

Thus Theorem 3.2 from Carmon and Duchi [2016] shows that with probability at least $1 - \delta'$ the iterates $\tilde{m}(\boldsymbol{\Delta}_t) \leq \tilde{m}(\boldsymbol{\Delta}^\star) + (1+\bar{\sigma})\epsilon' \leq \tilde{m}(\boldsymbol{\Delta}^\star) + 2\epsilon' = \tilde{m}(\boldsymbol{\Delta}^\star) + c_3\frac{\rho}{12}\|\boldsymbol{\Delta}_t^\star\|^3$ for $t \geq \mathcal{T}(\epsilon)$. We can now upper bound $\mathcal{T}(\epsilon)$. Recall we set $L = \frac{1}{2}\sqrt{\frac{\epsilon}{\rho}}$ for clarity.

$$\mathcal{T}(\epsilon) = \frac{1+\bar{\sigma}}{\eta} \min\left\{\frac{1}{\rho\|\boldsymbol{\Delta}^\star\| - \gamma}, \frac{10\|\boldsymbol{\Delta}^\star\|^2}{\epsilon'}\right\} \times$$

$$\left[6\log\left(1 + \mathbb{I}_{\{\gamma>0\}}\frac{3\sqrt{d}}{\bar{\sigma}\delta}\right) + 14\log\left(\frac{(\beta + \rho\|\boldsymbol{\Delta}^\star\|)\|\boldsymbol{\Delta}^\star\|^2}{\epsilon'}\right)\right]$$

$$\leq \mathcal{O}(1) \cdot \frac{\ell}{\rho\|\boldsymbol{\Delta}^\star\|} \times \left[6\log\left(1 + \frac{\sqrt{d}}{\delta'}\right) + 6\log\left(9\frac{\|\boldsymbol{\Delta}^\star\|^3}{L^3}\right) + 14\log\left(\frac{24\beta}{c_3\rho\|\boldsymbol{\Delta}^\star\|} + \frac{48}{c_4}\right)\right]$$

$$\leq \mathcal{O}(1) \cdot \frac{\ell}{\rho\|\boldsymbol{\Delta}^\star\|} \times \left[\mathcal{O}(1) \cdot \log\left(1 + \frac{\sqrt{d}}{\delta'}\right) + \mathcal{O}(1) \cdot \log\left(\frac{\|\boldsymbol{\Delta}^\star\|}{L}\right) + \mathcal{O}(1) \cdot \log\left(\frac{\beta}{\rho\|\boldsymbol{\Delta}^\star\|} + 1\right) + \mathcal{O}(1)\right]$$

$$\leq \mathcal{O}(1) \cdot \frac{\ell}{\rho\|\boldsymbol{\Delta}^\star\|} \times \left[\mathcal{O}(1) \cdot \log\left(1 + \frac{\sqrt{d}}{\delta'}\right) + \mathcal{O}(1) \cdot \log\left(\frac{\beta}{\rho L} + \frac{\|\boldsymbol{\Delta}^\star\|}{L}\right) + \mathcal{O}(1)\right]$$

$$\leq \mathcal{O}(1) \cdot \frac{\ell}{\rho^{\frac{1}{2}}\sqrt{\frac{\epsilon}{\rho}}} \times \left[\mathcal{O}(1) \cdot \log\left(1 + \frac{\sqrt{d}}{\delta'}\right) + \mathcal{O}(1) \cdot \log\left(\frac{\ell}{\rho^{\frac{1}{2}}\sqrt{\frac{\epsilon}{\rho}}} + \frac{\|\boldsymbol{\Delta}^\star\|}{\frac{1}{2}\sqrt{\frac{\epsilon}{\rho}}}\right) + \mathcal{O}(1)\right]$$

$$\leq \mathcal{O}(1) \cdot \ell \sqrt{\frac{1}{\rho\epsilon}} \times \left[ \mathcal{O}(1) \cdot \log\left(1 + \frac{\sqrt{d}}{\delta'}\right) + \mathcal{O}(1) \cdot \log\left(\frac{\ell}{\rho\sqrt{\frac{1}{\epsilon\rho}}} + \frac{\ell}{\rho\sqrt{\frac{1}{\epsilon\rho}}}\right) + \mathcal{O}(1) \right]$$

$$\leq \mathcal{O}(1) \cdot \ell \sqrt{\frac{1}{\rho\epsilon}} \times \left[ \mathcal{O}(1) \cdot \log\left(1 + \frac{\sqrt{d}}{\delta'}\right) + \mathcal{O}(1) \cdot \log\left(\ell\sqrt{\frac{1}{\rho\epsilon}}\right) + \mathcal{O}(1) \right]$$

$$\leq \mathcal{O}(1) \cdot \ell \sqrt{\frac{1}{\rho\epsilon}} \times \left[ \log\left(\ell\sqrt{\frac{1}{\rho\epsilon}}\left(1 + \frac{\sqrt{d}}{\delta'}\right)\right) + \mathcal{O}(1) \right],$$

where we have used the bound $\|\mathbf{\Delta}^\star\| \leq \mathcal{O}(1) \cdot \frac{\ell}{\rho}$. We can see that $\|\mathbf{\Delta}^\star\| \leq \mathcal{O}(1) \cdot \frac{\ell}{\rho}$ by appealing to the first-order stationary condition in Equation (11), $\|\mathbf{g}\| \leq \frac{\ell^2}{\rho}$, and $\|\mathbf{B}\mathbf{\Delta}^\star\| \leq (1 + c_2)\ell\|\mathbf{\Delta}^\star\|$. Combining these facts[6] we have:

$$\mathbf{g} + \mathbf{B}\mathbf{\Delta}^\star + \frac{\rho}{2}\|\mathbf{\Delta}^\star\|\mathbf{\Delta}^\star = 0 \implies$$

$$\frac{\rho}{2}\|\mathbf{\Delta}^\star\|^2 \leq \|\mathbf{g}\| + \|\mathbf{B}\mathbf{\Delta}^\star\| \leq \frac{\ell^2}{\rho} + (1 + c_2)\ell\|\mathbf{\Delta}^\star\| \leq \left(1 + (1 + c_2)^2\right)\frac{\ell^2}{\rho} + \frac{\rho}{4}\|\mathbf{\Delta}^\star\|^2 \implies$$

$$\|\mathbf{\Delta}^\star\| \leq 3\frac{\ell}{\rho},$$

for sufficiently small $c_2$. This completes the proof of the first statement of the Lemma.

Note that we will eventually choose $\delta' \sim \mathcal{O}(\frac{1}{\epsilon^{1.5}})$ for our final guarantee. However, this will only contribute logarithmic dependence in $\epsilon$ to our upper bound.

We now show that the iterate $\left\|\mathbf{\Delta}_{\mathcal{T}(\epsilon)}\right\| \leq \|\mathbf{\Delta}^\star\| + c_4\sqrt{\frac{\epsilon}{\rho}}$. For notational convenience let us use $\left\|\tilde{\mathbf{\Delta}}^\star\right\|$ to denote the norm of the global minima of the *perturbed* subproblem. Since our step size choice and initialization satisfy **Assumptions A** and **B** then we can also apply Corollary 2.5 from Carmon and Duchi [2016]. Corollary 2.5 states the norms $\|\mathbf{\Delta}_t\|$ are non-decreasing and satisfy $\|\mathbf{\Delta}_t\| \leq \left\|\tilde{\mathbf{\Delta}}^\star\right\|$. So we immediately obtain $\left\|\mathbf{\Delta}_{\mathcal{T}(\epsilon)}\right\| \leq \left\|\tilde{\mathbf{\Delta}}^\star\right\|$. We can then use Lemma 4.6 from Carmon and Duchi [2016] which relates the norm of the global minima of the *perturbed* subproblem, $\left\|\tilde{\mathbf{\Delta}}^\star\right\|$, to the norm of the global minima of the original problem, $\|\mathbf{\Delta}^\star\|$. Lemma 4.6 from Carmon and Duchi [2016] states that under the gradient perturbation $\tilde{\mathbf{g}} = \mathbf{g} + \sigma q$ we have that $\left|\left\|\tilde{\mathbf{\Delta}}^\star\right\|^2 - \|\mathbf{\Delta}^\star\|^2\right| \leq \frac{4\sigma}{\rho}$. So using the upper bound on $\sigma$ from Equation (27) we obtain that:

$$\left\|\mathbf{\Delta}_{\mathcal{T}(\epsilon)}\right\|^2 \leq \left\|\tilde{\mathbf{\Delta}}^\star\right\|^2 \leq \|\mathbf{\Delta}^\star\|^2 + \frac{4\sigma}{\rho} \leq \|\mathbf{\Delta}^\star\|^2 + \frac{c_3}{576}\frac{\epsilon}{\rho} \implies$$

$$\left\|\mathbf{\Delta}_{\mathcal{T}(\epsilon)}\right\| \leq \|\mathbf{\Delta}^\star\| + \sqrt{\frac{c_3}{576}}\sqrt{\frac{\epsilon}{\rho}} = \|\mathbf{\Delta}^\star\| + c_4\sqrt{\frac{\epsilon}{\rho}},$$

where we define $c_4 = \sqrt{\frac{c_3}{576}}$. $\qquad\square$

## C.3 Proofs of Lemma 1 and Corollary 1

Here we conclude by showing the correctness of Algorithm 2 which follows easily using our previous results.

**Lemma 1.** *There exists an absolute constant $c'$, such that under the same assumptions on $f(\mathbf{x})$ and the same choice of parameters $n_1, n_2$ as in Theorem 1, Algorithm 2 satisfies Condition 1 with probability at least $1 - \delta'$ with $\mathcal{T}(\epsilon) \leq \tilde{\mathcal{O}}(\frac{\ell}{\sqrt{\rho\epsilon}})$.*

*Proof.* This result follows immediately from Lemmas 7 and 8. In particular, Lemma 7 shows the Cauchy step, which is only used when $\|\mathbf{g}\| \geq \frac{\ell^2}{\rho}$, satisfies the conditions of an **Case 1** procedure.

Lemma 8 shows solving the cubic submodel via gradient descent, which is only used when $\|\mathbf{g}\| \le \frac{\ell^2}{\rho}$, satisfies the conditions of a **Case 2** procedure. Lemma 7 shows the Cauchy step has gradient complexity $\mathcal{O}(1)$. Lemma 8 shows the gradient complexity of the gradient descent loop is upper bounded by $\tilde{\mathcal{O}}(\frac{\ell}{\sqrt{\rho\epsilon}})$ but has a failure probability $1 - \delta'$ over the randomness in the gradient perturbation. $\qquad\square$

Finally, assembling all of our results we can conclude that: Using Lemma 1 and Theorem 1 we can immediately see that:

**Corollary 1.** *Under the same settings as Theorem 1, if $\epsilon \le \min\left\{\frac{\sigma_1^2}{c_1 M_1}, \frac{\sigma_2^4}{c_2^2 M_2^2 \rho}\right\}$, and if we instantiate the Cubic-Subsolver subroutine with Algorithm 2, then with probability greater than $1 - \delta$, Algorithm 1 will output an $\epsilon$-second-order stationary point of $f(\mathbf{x})$ within*

$$\tilde{\mathcal{O}}\left(\frac{\sqrt{\rho}\Delta_f}{\epsilon^{1.5}}\left(\frac{\sigma_1^2}{\epsilon^2} + \frac{\sigma_2^2}{\rho\epsilon} \cdot \frac{\ell}{\sqrt{\rho\epsilon}}\right)\right) \tag{7}$$

*total stochastic gradient and Hessian-vector product evaluations.*

*Proof.* We can check the descent constants (with respect to the cubic submodel) referenced in the proof of Theorem 1 for the **Case 1** and **Case 2** procedures of Algorithm 2, $K_1 = \frac{7}{20}$ and and $K_2 \le \frac{1}{96}$ respectively, satisfy $-K_1 \le -K_2$. The conclusion then follows immediately from Theorem 1, since Lemma 1 shows that Algorithm 2 is a Cubic-Subsolver routine, as defined in Condition 1, with iteration complexity $\mathcal{T}(\epsilon) \le \tilde{\mathcal{O}}(\frac{\ell}{\sqrt{\rho\epsilon}})$. $\qquad\square$

## D   Experimental Details

### D.1   Synthetic Nonconvex Problem

The W-shaped function used in our synthetic experiment is a piecewise cubic function defined in terms of a slope parameter $\epsilon$ and a length parameter $L$:

$$w(x) = \begin{cases} \sqrt{\epsilon}\left(x + (L+1)\sqrt{\epsilon}\right)^2 - \frac{1}{3}\left(x + (L+1)\sqrt{\epsilon}\right)^3 - \frac{1}{3}(3L+1)\epsilon^{3/2}, & x \le -L\sqrt{\epsilon}; \\ \epsilon x + \frac{\epsilon^{3/2}}{3}, & -L\sqrt{\epsilon} < x \le -\sqrt{\epsilon}; \\ -\sqrt{\epsilon}x^2 - \frac{x^3}{3}, & -\sqrt{\epsilon} < x \le 0; \\ -\sqrt{\epsilon}x^2 + \frac{x^3}{3}, & 0 < x \le \sqrt{\epsilon}; \\ -\epsilon x + \frac{\epsilon^{3/2}}{3}, & \sqrt{\epsilon} < x \le L\sqrt{\epsilon}; \\ \sqrt{\epsilon}\left(x - (L+1)\sqrt{\epsilon}\right)^2 + \frac{1}{3}\left(x - (L+1)\sqrt{\epsilon}\right)^3 - \frac{1}{3}(3L+1)\epsilon^{3/2}, & L\sqrt{\epsilon} \le x. \end{cases}$$

We set $\epsilon = 0.01$ and $L = 5$ in our experiment.

For stochastic cubic regularization, we fix $\rho = 1$ at the analytic Hessian Lipschitz constant for this problem, and we use 10 inner iterations for each invocation of the cubic subsolver, finding that this yields a good trade-off between progress and accuracy. Then for each method, we perform a grid search over the following hyperparameters:

- Batch size: $\{10, 30, 100, 300\}$
- Step size: $\{c \cdot 10^{-i} : c \in \{1, 3\}, i \in \{1, 2, 3, 4, 5\}\}$

Gradient and Hessian batch sizes are tuned separately for our method. We select the configuration for each method that converges to a global optimum the fastest, provided the objective value stays within 5% of the optimal value after convergence. Since the global optima are located at $(\pm\frac{3}{5}, 0)$ and each has objective value $-\frac{2}{375}$, this is equivalent to an absolute tolerance of $\frac{1}{3750} = 0.0002666\cdots$.

## D.2 Deep Autoencoder

Since the standard MNIST split does not include a validation set, we separate the original training set into 55,000 training images and 5,000 validation images, plotting training error on the former and using the latter to select hyperparameters for each method. We choose the configuration of each method that minimizes validation loss after a fixed budget of 2,000,000 oracle calls.

In addition to carrying out experiments using our method, SGD, and AdaGrad, we also test a hybrid variance reduction and second-order method using the framework of Reddi et al. [2017]. As proposed in their work, we use SVRG [Johnson and Zhang, 2013] for the GRADIENT-FOCUSED-OPTIMIZER. For the HESSIAN-FOCUSED-OPTIMIZER, we use Oja's method [Jain et al., 2016] applied to $I - \eta \nabla^2 f$ to compute an approximate minimum Hessian eigenvalue, then take a step in that direction.

Due to computational constraints, we were unable to perform a full grid search over all hyperparameters for every method. As a compromise, we fix all batch sizes and tune the remaining hyperparameters. In particular, we fix the gradient batch size for all methods at 100, a typical value in deep learning applications, and use a Hessian batch size of 10 for stochastic cubic regularization and the hybrid method as motivated by the theoretical scaling. Then we perform a grid search over step sizes for each method using the same set of values from the synthetic experiment. For stochastic cubic regularization, we also select $\rho$ from $\{0.01, 0.1, 1\}$ but find that the choice of this value had little effect on final performance. As in the synthetic experiments, we carry out 10 inner iterations per invocation of the cubic subsolver. For the hybrid method, we separately tune the step sizes of the SVRG phase and the Oja phase. We additionally select the number of iterations for SVRG from $\{100, 300, 1000, 3000\}$ and the number of iterations for Oja's method from $\{30, 100, 300, 1000\}$.

## Footnotes

[3]See Appendix Section C.2 for more details.

[4]We can also see with this rescaled $\epsilon'$ the step-size requirement in Lemma 6 will be satisfied.

[5]Note there is also a factor of 2 difference used in the Hessian-Lipschitz constant in Carmon and Duchi [2016]. Namely the $2\rho' = \rho$ for every $\rho'$ that appears in Carmon and Duchi [2016].

[6]The Fenchel-Young inequality $ab \leq a^2 + \frac{b^2}{4}$ is also used in the second line of this display.