[Reviews · NeurIPS 2018]

Reviewer 1



This submission is interested in stochastic nonconvex optimization problems, in which only stochastic estimates of the objective and its derivatives can be accessed at every iteration. The authors develop a variant of the cubic regularization framework, that only requires access to stochastic gradients and products of stochastic Hessians with vectors. Such a method is shown to reach a point at which the gradient norm is smaller than $\epsilon$ and the minimum Hessian eigenvalue is bigger than $-\sqrt{\rho \epsilon}$ in a number of stochastic queries (gradient or Hessian-vector product) of order of $\epsilon^{-3.5}$, which improves over the classical complexity of Stochastic Gradient Descent (SGD). The problem of interest is clearly introduced, along with the possible advantages of using both stochastic estimates and a cubic regularization framework. The associated literature is correctly reviewed, and the authors even cite contemporary work that achieves similar complexity guarantees but rely on stochastic gradient estimates and variance reduction. It seems that these results could also have appeared in the table (Table 1) provided by the authors. Section 3 of the submission contains its main results and a meta-algorithm that achieves the desired complexity rate, while Section 4 sketches the proof of the main result. The description of the algorithm and the properties of its subroutines are clearly presented. I would like to point out that the main algorithm resembles Algorithm 1 of [Carmon et Duchi, 2016], especially regarding the final application of the cubic subsolver: since the authors cite this reference, it seems that the algorithmic process cannot be considered as completely original. Section 5 presents empirical results for the performance of the method, compared to two variants of SGD. The results are promising for the proposed algorithm, and several details about those experiments can be found in appendix. Those tests are undoubtedly relevant, however, one could also have expected a comparison with at least one variance-reduction method achieving a similar complexity bound, since those seem to be the only competitors on the theoretical side. I am thus surprised that the authors do not comment on such testing. Overall, it is my opinion that this submission presents original and interesting results, that complement the existing landscape of algorithms in a meaningful way, They propose a method that relies on stochastic estimates together with an inexact solve of the cubic subproblem: their method is thus particularly appealing, although it is unclear whether this particular algorithm will outperform others possessing similar complexity guarantees. The submission is clearly written, and I particularly appreciate the balance between the main body of the submission and its appendices. As a result, I will give this submission an overall score of 8. Additional comments: a) There appears to be several articles and conjunctions missing throughout the submission (e.g., "the" missing in "We also make following assumptions" L.117; "that" missing in "this will show if the cubic submodel" L.497). b) In Table 1 page 3, I believe the footnote "1" on the bound for Natasha 2 should actually be a "2". c) On Line 8 of Algorithm 2, I believe there is an extra $\eta$ in front of the $B[\Delta]$ term. d) In Appendix E.1, it is said that a grid search was run over the hyperparameters of each method, but no information about the final configuration is provided. In particular, and contrary to Appendix E.2, the final adopted sample size is not given, yet this would be a relevant information for this submission. Edit following author feedback: I would like to point out that my comment on Appendix E was in favor of more details in Appendix E.1, not E.2 (in Appendix E.2, the final choice for gradient and Hessian batch sizes is clearly stated; this is not the case in Appendix E.1, where a range of options is given without specifying which option is the one used in the experiments). Apart from this misunderstanding, I am satisfied by the way the authors addressed my comments. I trust them to follow up on their response and incorporate a comparison with other second-order methods in their revised version, as this was also pointed out by another referee. Since all reviewers agreed on the overall score to give to this submission, I will maintain my initial assessment of the submission.

Reviewer 2



This paper proposes a stochastic cubic regularized Newton method to escape saddle points of non-convex functions. In particular, it explores the second-order information to escape the saddle points by utilizing the cubic regularization. The convergence rate of the proposed method can match the existing best-known result. Overall, this paper is well structured and presented clearly. I have some concerns as follows: 1. In the experiment, is the employed SGD the classical SGD? If not, it’s better to compare the proposed method with the variant proposed in [1]. 2. Since the proposed method utilizes the second-order information. It’s better to compare it with the second-order counterparts, such as Natasha2. [1]. Ge, Rong, et al. "Escaping from saddle points—online stochastic gradient for tensor decomposition." Conference on Learning Theory. 2015. After feedback: Since the purpose of this algorithm is to escape the saddle point, it is recommended to compare it with the perturbed SGD, which is designed to escape saddle points, in your final version.

Reviewer 3



The paper analyzes a stochastic algorithm which builds on the cubic-regularized Newton method. The algorithm uses gradient and Hessian-vector product evaluations and improves over existing techniques in escaping sadlles and finding local minima. Experiments are run on both synthetic and a real-world problem, and compared with SGD. The authors provide theoretical guarantees. This is a solid incremental paper, which is worth accepting (in my opinion). Other manuscripts provide almost equivalent results. After feedback: the authors appear to fully understand and agree on the need of further clarifications on the numerics: 1) the comparison to (at least) one other variance-reduction method will be provided 2) the hyperparameter tuning process will be critically displayed Moreover, the authors are going to add information on the comparison with competing results. Overall, the author appear to agree with all reviewers on the potential weakness of the paper and their reply is fully satisfactory. The numerical experiments corroborate nicely the overall framework. The rigorous guarantees remain the strong strand of the paper.